# Protease-mediated activation of Par2 elicits calcium waves during zebrafish egg activation and blastomere cleavage

Jiajia Ma[iD], Tom J. Carney[iD]*

Lee Kong Chian School of Medicine, Yunnan Garden Campus, Nanyang Technological University, Singapore, Singapore

* tcarney@ntu.edu.sg

## Abstract

Successful initiation of animal development requires activation of the egg immediately prior to fusion of gamete pronuclei. In all taxa, this is initiated by waves of calcium transients which transverse across the egg. Calcium waves also occur at cleavage furrows during later blastula cytokinesis. Calcium is released from the endoplasmic reticulum through activation of inositol-1,4,5-trisphosphate ($IP_3$) receptors. Only a subset of the mechanisms employed to generate $IP_3$ during vertebrate egg activation are defined, with strong evidence that other critical mechanisms exist. Serine proteases have been long implicated in egg activation and fertilization. Here, we report that treatment of zebrafish eggs with serine protease inhibitors leads to defective calcium wave propagation and failed egg activation. We further show that mutation of zebrafish Protease-activated receptor 2a (Par2a) also results in severe disruption of egg activation, leading to failed chorion elevation and ooplasmic segregation. Milder *par2a* mutants progress further, but then show abnormal blastomere cleavage. We observed that *par2a* mutants show decreased amplitude and duration of calcium transients. Restoring $Ca^{++}$ or direct injection of $IP_3$ ligand rescues egg activation aborted by either serine protease inhibitor treatment or by mutation of Par2a. We thus show that serine protease activity is a critical regulator of $IP_3$ and subsequent calcium wave amplification during zebrafish egg activation, and link this to intracellular calcium release via the protease receptor, Par2a. This constitutes a novel signaling pathway critical for successful fertilization.

## Introduction

Fertilization culminates in the fusion of sperm and egg pronuclei to generate a zygote. Prior to this, both gametes must undergo a number of maturation and activation steps to achieve fertilization competence [1]. The final activation process of the oocyte is termed egg activation, which occurs upon, or immediately prior to,

**Data availability statement:** The raw sequencing data for the RNA-seq analysis is deposited in NCBI's Gene Expression Omnibus, GEO Series accession number GSE289416 (https://www.ncbi.nlm.nih.gov/geo/query/acc.cgi?acc=GSE289416).

**Funding:** This work was funded by a Ministry of Education (MoE) Academic Research Fund (AcRF) Tier 1 grant 2018-T1-001-184 to TJC. The funders had no role in study design, data collection and analysis, decision to publish, or preparation of the manuscript. This Tier 1 AcRF grant included salary for JM.

**Competing interests:** The authors have declared that no competing interests exist.

**Abbreviations:** BSA, Bovine Serum Albumin; CGE, Cortical Granule Exocytosis; CGs, cortical granules; FITC-MPL, FITC-conjugated *Maclura pomifera* Lectin; GPCRs, G protein-coupled receptors; IP$_3$, inositol-1,4,5-trisphosphate; Par2a, Protease-activated receptor 2a; PLC, Phospholipase C; PLCζ, phospholipase C zeta; SOCE, store-operated calcium entry; 2-APB, 2-aminoethyl diphenyl borinate.

fertilization. Egg activation has broad conservation across animals and achieves largely comparable outcomes in all species [2]. These include exocytosis of cortical granules (CGs), the cortical reaction, formation of the pronucleus, resumption of meiosis and extrusion of the second polar body. These are essential for blocking polyspermy and for subsequent development to proceed. In fish species, the release of CGs during activation elevates the chorion and creates a perivitelline fluid filled space between the chorion and the egg. There is also a significant reorganization of the egg cytoplasm which is initially intermingled with lipid rich yolk granules in arrested fish eggs. Upon activation, actin-myosin contractions squeeze this ooplasm away from the yolk to the animal pole where it forms the blastodisc [3,4].

In mammals, egg activation is initiated by binding of sperm to the egg during the process of internal fertilization. In most teleost fish, fertilization occurs externally by broadcast spawning. For example, during mating courtship, zebrafish males wrap around the body of the female and squeeze the abdomen, releasing the eggs into the water. Simultaneously the male releases sperm [5]. Although release of sperm and eggs into the water are spatially and temporally coincident, the factor activating zebrafish eggs is contact with water. It does not require fertilization by sperm and can occur parthenogenetically [6,7]. Thus, egg activation and fertilization are separate processes but occur almost simultaneously during natural mating so that egg activation timing promotes monospermic fertilization. Prior to spawning, the ovary appears to have properties blocking precocious egg activation, and spontaneous activation of zebrafish eggs following release into water can be blocked by incubation in fish ovarian fluid [8]. The critical factor imparting this property has been proposed to be a serpin-type protease inhibitor which is found at high levels in zebrafish ovarian fluid [9]. Furthermore, treatment of loach and carp eggs with protease inhibitors also prevented spontaneous activation [10], although it is unclear how protease inhibitors block egg activation.

Intracellular calcium has a fundamental and conserved role in driving egg activation, fertilization, and early embryo development across both vertebrates and invertebrates [2,7,11–13]. The dynamics of calcium varies in the eggs of different species. In most invertebrates, fish, and amphibia, egg activation is driven by a single $Ca^{++}$ wave that transverses the egg, while in mammals, fertilization initiates multiple oscillatory waves [6,7]. Irrespective of initiation mode or $Ca^{++}$ wave pattern, assays in multiple species including sea urchins, *Xenopus*, fish, and mammals, have demonstrated the second messenger, inositol-1,4,5-trisphosphate (IP$_3$) releases intracellular $Ca^{++}$ in eggs via binding to IP$_3$ receptors on the endoplasmic reticulum. These IP$_3$-generated $Ca^{++}$ waves are sufficient to activate eggs [2,14–19].

In amphibia and fish, distinct $Ca^{++}$ transients are later observed extending laterally along the nascent division plane of cleavage furrows prior to blastomere cytokinesis [20–22]. These transient waves of $Ca^{++}$ are associated with furrow positioning, propagation, and deepening [23]. As with egg activation, there is a requirement for IP$_3$ in these transients although there is evidence that each $Ca^{++}$ transient may have distinct regulation mechanics including a need for cytosolic $Ca^{++}$ replenishment via store-operated calcium entry (SOCE) [24–26].

The varied mechanisms and environments initiating egg activation across the animal kingdom and the diverse patterns of $Ca^{++}$ waves employed therein and subsequently during blastomere cleavage indicates that there are highly likely to be multiple $Ca^{++}$ regulating mechanisms. As $IP_3$ is generated by cleavage of $PIP_2$ through the activity of Phospholipase C (PLC) family members, it is likely that PLC regulation plays a central role in egg activation. The first $Ca^{++}$ transient in mouse is initiated by the release of phospholipase C zeta (PLCζ) from the sperm into the egg cytoplasm [27]. In sea urchin and xenopus, FAK and Src-family protein tyrosine kinases contribute to $Ca^{++}$ release during egg activation through activation of PLCγ, although Src family kinases are entirely dispensable for $Ca^{++}$ release in the mouse and have only a partial effect in zebrafish [28–32].

In addition to kinases, PLCs are also activated by members of the heterotrimeric G protein alpha subunit Gq family (GNAQ, GNA11, GNA14, GNA15) which in turn are activated by certain G protein-coupled receptors (GPCRs). Protease Activated Receptors are a family of GPCRs activated by extracellular protease activity, through unmasking of a tethered ligand which binds to the receptor intra-molecularly, activating downstream signaling [33]. For example, Par2 (encoded by the *F2rl1* gene) is activated by trypsin-like proteases and has been shown to activate G alpha-q and in turn PLCβ, thus releasing calcium in a number of cell types and contexts [34–36].

Here, we show that zebrafish egg activation is sensitive to serine protease inhibitors, and identify that maternal mutants of the serine protease responsive receptor, Par2a, have broad egg activation defects due to altered calcium waves. This identifies a novel essential regulator of calcium during the activation of vertebrate eggs.

## Results

### Serine protease inhibition stalls egg activation

We wanted to determine if zebrafish egg activation was sensitive to protease inhibitors as shown for the loach and carp [10]. Eggs were squeezed from WT zebrafish females into Hank's Solution containing sperm. This solution also contained 0.5% BSA, which is known to hold eggs in an inactivated state and block fertilization [37]. Eggs were then activated by diluting out of the BSA through addition of excess E2 medium, thus permitting fertilization to proceed. Eggs showed immediate activation, rapidly raising chorions and displaying a prominent blastodisc after 30 min, with over 90% subsequently showing normal cell division indicating successful fertilization (Fig 1A). The addition of 5 mg/ml of the peptide-based serine protease inhibitor, Aprotinin, to the E2 medium, however, resulted in all eggs remaining fully inactivated even after 30mins, showing neither chorion elevation nor blastodisc formation and thus no cell division (100%, *n* > 200 eggs; Fig 1B). The presence of sperm made no difference to egg activation block by Aprotinin, indicating that Aprotinin is disrupting egg activation and not fertilization.

### Serine proteases activate eggs via $IP_3$-mediated calcium release

To determine if protease inhibition altered calcium dynamics during egg activation, we employed the *Tg(actb2:G-CaMP6s)^{lkc2}* transgenic line [35,38]. The β-*actin* promoter is active in oocytes and thus the GCaMP6s reporter protein is deposited in eggs. Following activation by exposure to E2 medium alone or by in vitro fertilization, WT eggs show rapid $Ca^{++}$ waves transversing the egg within 1 min. As noted by others, this occurred with comparable timing and extent, irrespective of fertilization [6] (S1 Movie). We then activated eggs from this line in E2 with or without Aprotinin and visualized fluorescence. $Ca^{++}$ transients in Aprotinin-treated eggs were shorter and less pronounced compared to E2-treated controls (Fig 1C and 1D and S2 Movie). Quantification over the first 10 min following Aprotinin treatment showed a significant reduction in amplitude and duration compared to E2-activated eggs (Fig 1E and 1F). Similar results were obtained using Benzamidine HCl, a small molecule inhibitor of trypsin-like serine proteases. Benzamidine reduced egg activation during in vitro fertilization, although less potently than Aprotinin, with eggs showing partial chorion lifting and small blastodisc formation (Figs 1G, 1G′, 1H, 1H′ and S1A, S1B). Benzamidine also significantly reduced duration of $Ca^{++}$ transients during

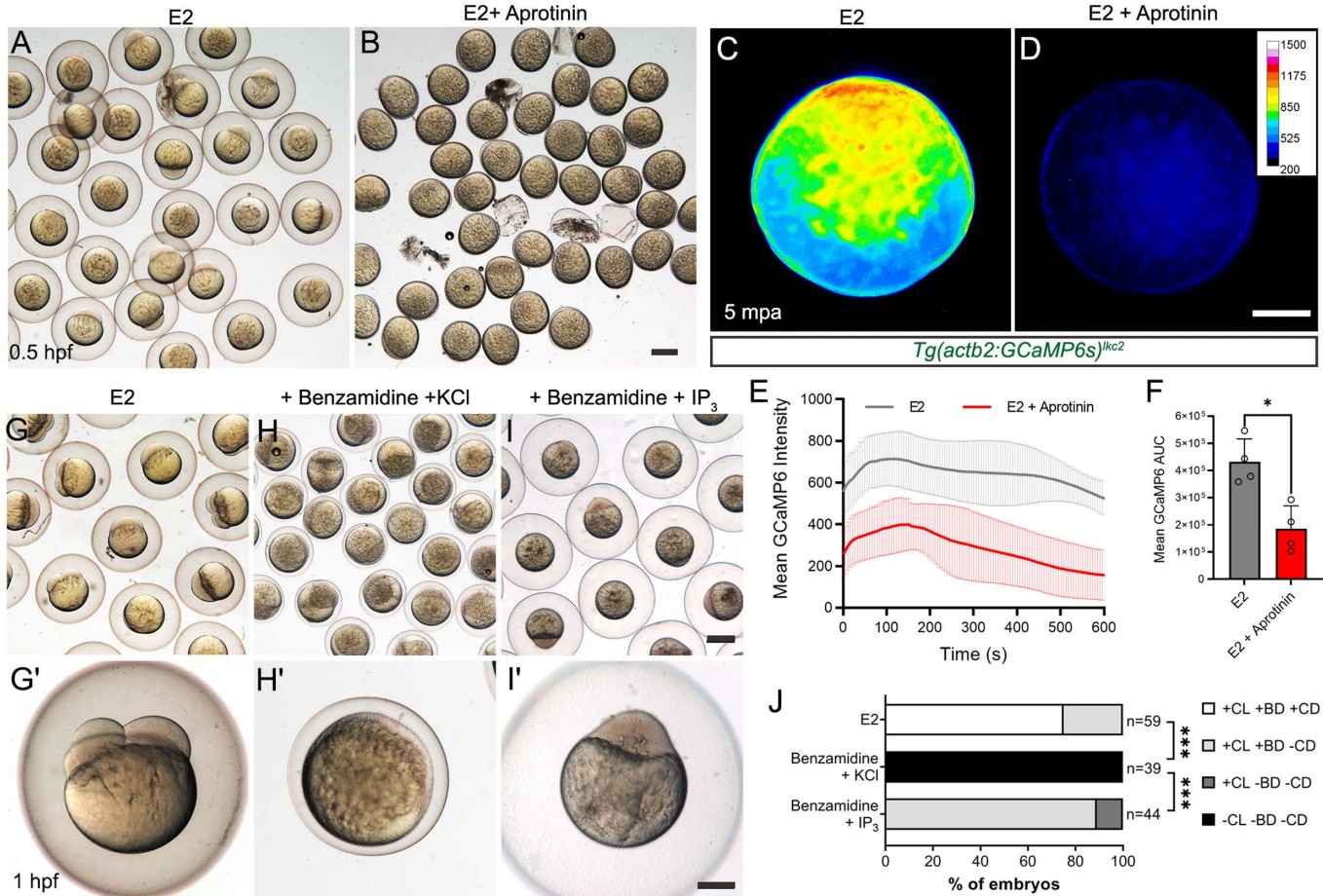

**Fig 1. Serine protease inhibitors cause egg activation defects and reduced calcium waves. A, B:** Eggs fertilized *in vitro* in E2 medium with (B) or without (A) 5 mg/ml Aprotinin. **C, D:** Normalized pseudo-coloured fluorescent images of eggs harvested from a *Tg(actb2:GCaMP6s)$^{lkc2}$* female, indicating Ca$^{++}$ levels at 5 min postactivation (mpa) in E2 (C) or in 5 mg/ml Aprotinin (D). **E:** Changes in GCaMP6s intensity over 10 min of *Tg(actb2:GCaMP6s)$^{lkc2}$* eggs activated with E2 (black) or 5 mg/ml Aprotinin (red). **F:** Corresponding Area Under Curve of GCaMP6 intensity levels presented in (E) following egg activation. $n=4$; Mann–Whitney test; * = $p < 0.05$. **G–I':** Low (G, H, I) and high (G', H, I') magnification images of embryos fertilized *in vitro* in E2 (G–G'), or in E2 with 40 mg/ml Benzamidine HCl treatment and subsequently injected with 0.7 mM KCl (H–H'), or in E2 with 40 mg/ml Benzamidine HCl treatment and injected with 10 mM IP$_3$ in KCl (I–I'). **J:** Proportion of embryos showing egg activation phenotypes and blastomere division. Benzamidine incubation aborts egg activation, which can be reactivated by IP$_3$ injection. Key: CL: Chorion Lift, BD: Blastodisc, CD: Cell Division, +: Present, −: Absent. Chi-squared analyses; *** = $p<0.001$. Scale bars: B, I = 500 μm; D, I' = 200 μm. See file S1 Data for underlying data.

egg activation compared to controls (S1C–S1E Fig and S3 Movie). Thus, the activity of serine proteases, most likely trypsin-like serine proteases, is necessary for the generation of normal calcium waves and egg activation in zebrafish.

Injection of IP$_3$ can activate eggs of many species [2]. We asked if the addition of IP$_3$ could reactivate eggs arrested by serine protease inhibition. We injected 20 pmol of IP$_3$ into eggs treated with Benzamidine and observed robust restoration of egg activation, with successful chorion lifting and formation of a blastodisc compared to Benzamidine-treated eggs injected with KCl injection solution only (Fig 1G, 1G', 1H, 1H', 1I, 1I' and 1J). However, these eggs were unable to undergo subsequent cell division, despite being derived from IVF, suggesting a serine protease is required for subsequent cell division in a process independent of IP$_3$. Thus, serine protease activity is required for egg activation through a Ca$^{++}$, and likely an IP$_3$-dependant process.

## Maternal *par2a* mutants display defects in egg activation and blastomere cleavage

How IP$_3$ is generated in eggs to initiate activation appears to be species-specific. It is unclear what generates IP$_3$ and how this may involve proteases. In zebrafish, Par2b has been implicated in IP$_3$ formation and intracellular Ca$^{++}$ release in the epidermis following activation by the serine protease Matriptase1a [35,39]. As *par2a* and *par2b* are also both maternally expressed [40], we hypothesized that they may link serine protease activity with IP$_3$ and Ca$^{++}$ generation during egg activation.

We generated *par2a* (= *f2rl1.1*) and *par2b* (= *f2rl1.2*) mutants through CRISPR/Cas9 targeting. CRISPR guide RNA was designed to target the second extracellular loop of the receptor. In all alleles, indels led to a frameshift leading to premature protein termination before the fifth transmembrane domain, and before the third intracellular and extracellular loops. The predicted protein in all alleles thus also lacks the C-terminal tail (S2A–S2C Fig). Quantitative PCR demonstrated a significant reduction of *par2a* mRNA levels in 1-cell zygotes derived from *par2a* mutant mothers suggesting nonsense-mediated decay of the mutant transcript (S3 Fig). Zygotic homozygous mutants for both *par2a* alleles, *par2a^lkc4, par2a^lkc6* (S2A Fig), were adult viable with no overt phenotype. Male homozygous *par2a* mutants were fertile and generated normal embryos when crossed to wild-type females (S1 Table). Female homozygous *par2a* mutant adults, however, gave rise to eggs that displayed a defect in activation, reminiscent of the activation phenotype seen in eggs treated with serine protease inhibitors. Thus, both eggs activated in vitro without sperm and embryos derived from natural spawning of *par2a^−/−* females with WT males showed significantly impaired chorion elevation and defective blastodisc formation compared to WT crosses or activated WT eggs (Fig 2A and S4 Movie and S1 Table). This indicated that the inability to undergo fertilization per se is not the primary defect in eggs lacking *par2a*. We noted that there was no statistical difference in clutch sizes generated by heterozygous *par2a* females, or *par2a* mutant females, suggesting there was no defect in oocyte/egg formation.

The expressivity of the egg activation phenotype in *par2a* mutants was variable. Severe mutant embryos showed only minimal chorion elevation and no blastodisc formation, while milder mutant embryos displayed only moderate chorion lifting and partial blastodisc formation, however, this blastodisc was often unstable and showed regions of partial collapse (Figs 2A and S4D). The effects are not due to simple delay, as even by 1.5 h postactivation, severely affected eggs remain with limited chorion elevation and no blastodisc (S4A–S4F Fig and S4 Movie). In more mildly affected fertilized eggs, some blastodiscs do form of sufficient size that they initiate blastomere cleavages, however resulting daughter cells are of unequal sizes, ultimately leading to failure to form normal blastomere tiers. (S5A and S5B Fig and S4 Movie). Blastomeres show reduced compaction and adherence to each other, and some detach into the perivitelline space (S5A and S5B Fig). All detached cells appear to retain nuclei (S5C Fig). Although mild *par2a* maternal mutant embryos are able to generate blastomeres, less than 1% successfully complete gastrulation, and those that do have severely affected axes and do not survive beyond 5dpf (S1 Table). Both alleles were identical in ability to generate severe or mild phenotypes. We found that the generation of mild or severe phenotypes was a property of each mother, not of each allele. Thus, different females of the same allele will consistently generate either severe or mildly affected clutches and there was a consistent severity within clutches (S5D Fig and S2 Table). Clutches from adult mutant sisters are also variable in phenotype expressivity, and we have not yet seen a drift of expressivity patterns across generations. To test if this variability was due to compensation by *par2b*, we made a *par2b* CRISPR allele (*par2b^lkc7*) which created a 2 bp deletion and 1 bp insertion indel leading to the generation of a premature stop codon and termination of the protein before the fifth transmembrane domain (S2B and S2C Fig). This allele of *par2b* was homozygous viable, and eggs from mutant females developed normally. As they are closely linked, to generate double mutants, we injected the *par2a* CRISPR into the *par2b* mutant background to create a third *par2a* allele, *par2a^lkc5*. Double *par2a; par2b* mutant females behaved the same as the *par2a* single mutants. They still spawned eggs with variable expressivity in egg activation phenotype, with no additive severity induced by combined loss of both paralogues (S1 Table). This indicates that compensation by *par2b* does not account for variable expressivity, and that *par2a*, but not *par2b*, is crucial for egg-to-embryo transition.

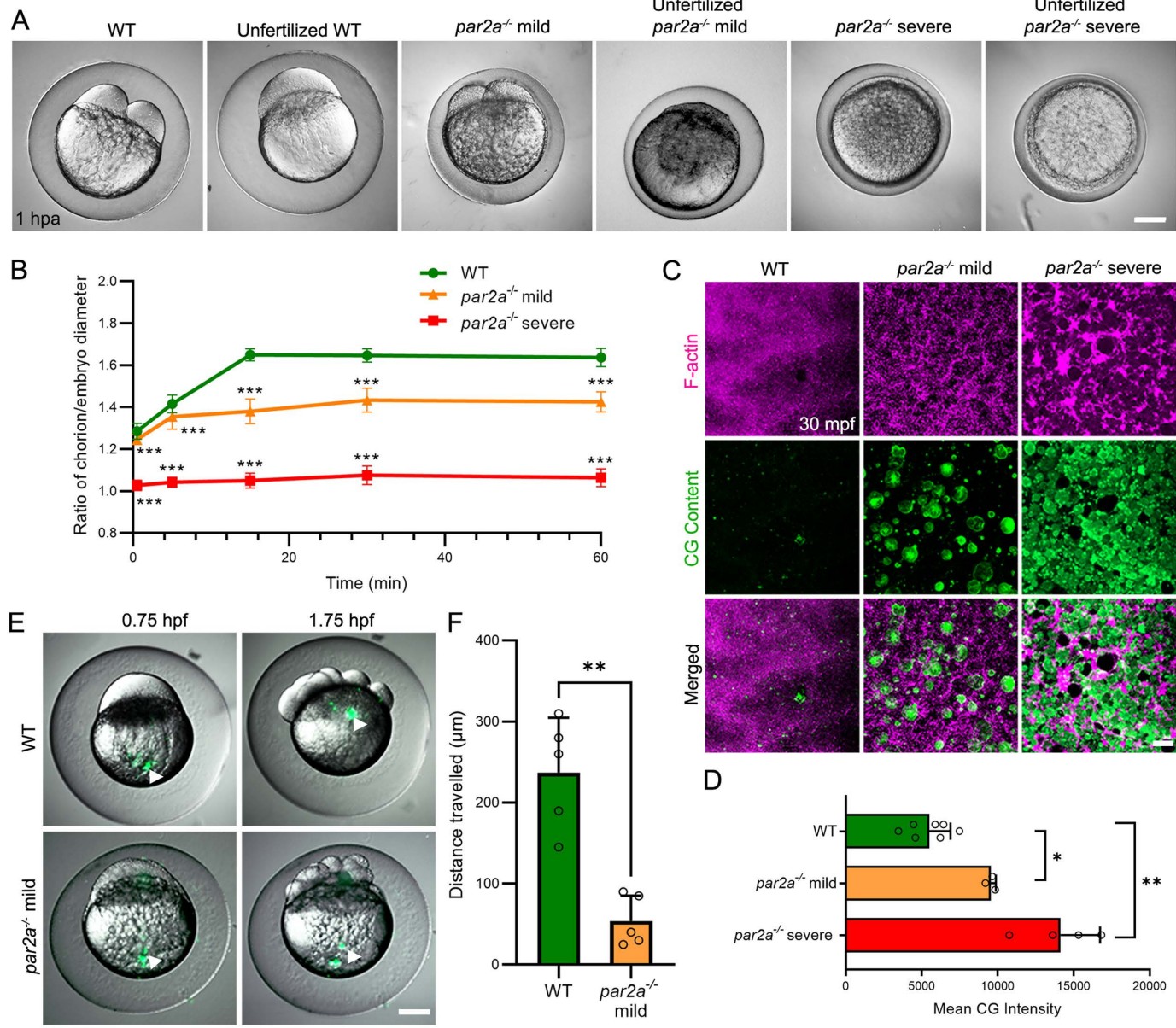

**Fig 2. Maternal *par2a* mutant embryos display impaired egg activation events and cell division defects. A:** Nomarski images of naturally fertilized and unfertilized, activated eggs from WT, mild and severe *par2a* mutant females. Chorion elevation and blastodisc formation are variably disrupted. **B:** Changes in the ratio of chorion lift to embryo size of naturally fertilized WT, mild and severe *par2a* mutant eggs at different time points over an hour postactivation, n=51; t-test; *** = *p*<0.001. **C**: Confocal images of F-actin (magenta) and cortical granule (CG Content; green) distribution in naturally fertilized WT (left), *par2a* mild (middle), and *par2a* severe (right) mutant eggs at 30 mpf. *par2a* mutants show retention of F-actin meshwork that correlates with retention of CGs at cortex. **D**: Quantification of CG content staining at cortex in fertilized WT, mild and severe *par2a* mutant eggs at 30 mpf. *n* = 7, 3, 4; Mann–Whitney test; * = *p*<0.05; ** = *p*<0.01. **E**: Fluorescent eGFP superimposed on Nomarski image of naturally fertilized WT and mild *par2a* mutants injected with fluorescent latex beads at the vegetal pole (left column). Bead movement is recorded 1 h later (right column). Arrowheads highlight latex bead position. **F**: Average distance traveled by injected fluorescent latex beads over 1 h in naturally fertilized WT and mild *par2a* mutant eggs. *n* = 5; Mann–Whitney test; ** = *p*<0.001. Scale bars: A, E = 200 μm; C = 20 μm. See file S1 Data for underlying data.

PLOS Biology

## Cortical granule release and cytoplasmic streaming fail in *par2a* mutants

We quantified the hallmarks of egg activation in mild and severe clutches. The ratio of diameters of the chorion to egg body was significantly reduced in both mild and severe *par2a* mutants compared to WT, with severe mutants having chorion diameter only slightly larger than the egg body after 60 min postfertilization (Fig 2A and 2B). Chorion elevation is driven by Cortical Granule Exocytosis (CGE), when CGs docked at the inner surface of the cortex, fuse with the plasma membrane and release their contents into the perivitelline space. This inflates the chorion outwards as well as hardens it [3]. The process is accompanied by cortical actin rearrangement of the egg surface within 5 min postegg activation (mpa), where the local network of F-actin is disassembled [3,41]. To determine if impaired chorion elevation in *par2a* mutants is due to defective CGE, we examined cortical actin distribution using AlexaFluor-546 Phalloidin. At 30 s after activation, WT eggs show a meshwork of F-actin under the plasma membrane. A similar meshwork is noted in activated eggs of both mild and severe *par2a* mutants (S5E Fig). In WT this rapidly reorganizes to become a smooth distribution of actin staining by 5 mpa. However, *par2a* mutant eggs fail to show this reorganization and retain an intensely stained meshwork appearance (S5E Fig). Even by 30-min postactivation, fertilized mild and severe *par2a* eggs retained this cortical actin meshwork, while it was disassembled in WTs (Fig 2C). As the actin meshwork is thought to act as a barrier to CGE, we directly stained CGs using FITC-conjugated *Maclura pomifera* Lectin (FITC-MPL). We observed that in contrast to WT which had almost complete release of CGs by 5 mpa, *par2a* mutants retained CGs at the egg surface even at 30 mpf (Figs 2C and S5F). By comparing mean FITC-MPL intensity in WT and *par2a* mutants at 30 mpa, we saw a significant increase in retention of CGs in mild mutants, and stronger retention in severe mutants (Fig 2D). This correlates with the severity of chorion elevation defect observed between severe and mild *par2a* mutants. Thus, we conclude that cortical F-actin disassembly and CGE fails in *par2a* mutant eggs, leading to defective chorion elevation.

After egg activation initiation, cytoplasm in the egg separates away from the yolk and streams towards the animal pole, which eventually becomes the blastodisc [4]. Because *par2a* mutants show defective blastodisc formation, we tested if cytoplasmic streaming is impaired by injecting fluorescent latex beads at the vegetal pole and tracking their movement over 1 h. We observed that in naturally fertilized WT eggs, most of the beads migrated to the animal pole, sitting at the forming blastodisc, however in fertilized *par2a* mutant eggs, injected beads showed little displacement (Fig 2E and S5 Movie). Quantification of latex bead displacement showed a significant reduction in *par2a* mutants compared to WT (Figs 2F and S5G). Together, these results implicate a novel role of Par2a in egg activation and subsequent blastomere cleavage.

## Ca$^{++}$ wave propagation is reduced in *par2a* mutants during egg activation and cell cleavage

To investigate if the defects observed in *par2a* mutants are Ca$^{++}$ dependent as seen for serine protease inhibitors, we crossed this mutant to the *Tg(actb2:GCaMP6s)$^{lkc2}$* transgenic line. Eggs from homozygous *par2a* mutant females were collected and GCaMP6s signals recorded by fluorescent time-lapse imaging following E2 activation. This revealed that compared to WT, *par2a* mutant eggs showed attenuation of the Ca$^{++}$ wave after activation either by fertilization or parthenogenetically. In all mutants, an initial focus of Ca$^{++}$ was seen, but this showed reduced intensity and duration, and failed to propagate as extensively as in WT. Extent of attenuation correlated with severity of subsequent phenotype, where embryos with most reduction in chorion lift and blastodisc formation had the smallest Ca$^{++}$ wave (Fig 3A and 3B and S6 and S7 Movies). Levels of calcium were reduced from the outset and remained so for the duration of imaging. Area under the curve analysis indicated significant attenuation of Ca$^{++}$ signals for both severe and mild *par2a* mutants (Fig 3C). Time-lapse analysis suggested that in severely affected eggs, there was an initial puff of calcium, however, this did not progress to a propagated wave (S7 Movie).

As mildly affected *par2a* mutant eggs form blastomeres, we were able to examine if the defects in blastomere cytokinesis were associated with disruptions in normal Ca$^{++}$ dynamics at the cleavage furrow. Characteristic, dynamic Ca$^{++}$ waves at the furrow were apparent in WT, with transient slow calcium waves at the positioning of each furrow (initiation wave),

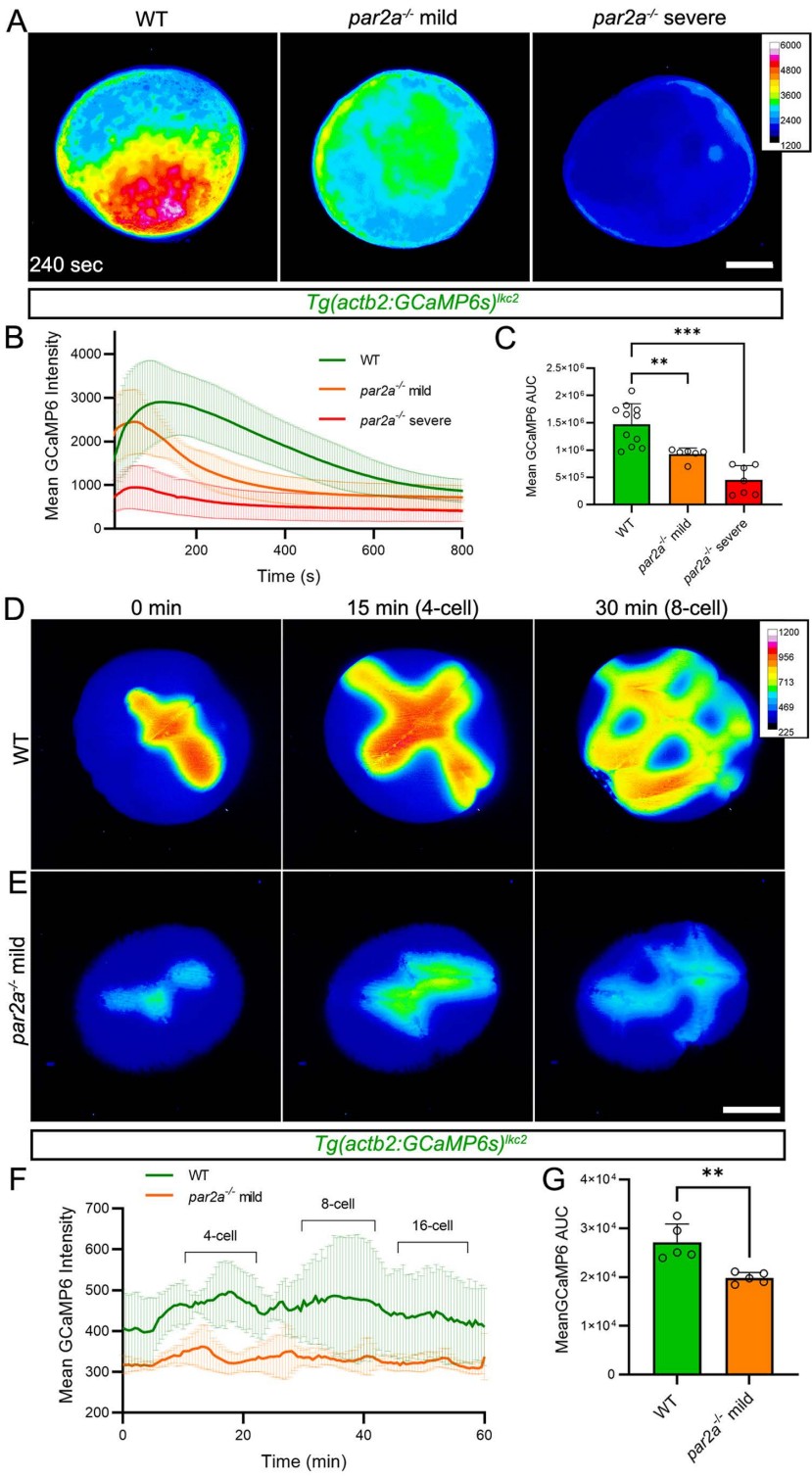

**Fig 3. *par2a* mutants show defective Ca++ wave propagation during egg activation and blastomere cytokinesis. A** Normalized pseudo-coloured fluorescent images of *Tg(actb2:GCaMP6s)^{lkc2}* indicating comparative Ca++ levels at 240 s postegg activation in WT, mild and severe *par2a* mutant eggs. **B:** Changes in GCaMP6s intensity over 10 min from the *Tg(actb2:GCaMP6s)^{lkc2}* Ca++ reporter transgene of WT (green), mild *par2a* (yellow) and severe

*par2a* (red) mutants following egg activation. **C:** Graph of corresponding Area Under Curve analysis of GCaMP6s levels presented in (B). $n = 11,6,7$; Mann–Whitney test; ** $= P < 0.01$; *** $= p < 0.001$. **D, E:** Projected light-sheet images of the animal pole of *Tg(actb2:GCaMP6s)*[lkc2] in naturally fertilized WT (D) and mild *par2a* mutant eggs (E) during cell division over 30 min starting from 2-cell stage. **F:** Changes in GCaMP6s intensity over 1 h in WT (green) and mild *par2a* mutant embryos (yellow) during early blastomere division. Acquisition begins at 20 mpf and cell division events are indicated above WT by brackets. **G:** Graph of corresponding Area Under Curve analysis of GCaMP6s levels presented in (F). $n = 5$; Mann–Whitney test; ** $= p < 0.01$. Scale bars: A, E $= 200\ \mu m$. See file S1 Data for underlying data.

which are then propagated as maturation waves along the furrow during deepening and zippering phases (Fig 3D and S8 Movie) [20,38]. Although there were weak transients at the nascent furrows of *par2a* mutants (likely initiation waves), these failed to propagate effectively, failing to fully extend laterally across the division plane (Fig 3E and S8 Movie). This coincided with loss of the orthogonal arrangement of cleavages in these embryos. Quantifying intensity of calcium transients during early blastomere cleavages showed a significant and sustained reduction in the mutants (Figs 3F and 3G). These experiments indicate that *par2a* is required for intensity and propagation of calcium waves during egg activation and at cleavage furrows during division of the blastomeres. To observe Par2a localization in the dividing blastomeres, we expressed a Par2a-eGFP fusion protein in the early zygote by use of the maternally active *zp3* promoter. As expected, Par2a-eGFP signal was observed at the cleavage furrows and cell membranes (S9 Movie).

### Defects in *par2a* mutants are rescued by increasing intracellular Ca$^{++}$

To test if this loss of Ca$^{++}$ indeed does account for the egg activation and cell division defects in *par2a* mutants, we asked if experimentally increasing intracellular Ca$^{++}$ would rescue any aspects of the phenotype. The ionophore, Ionomycin, raises intracellular Ca$^{++}$ in zebrafish eggs leading to activation [11]. 2 μM of ionomycin was added to eggs from a severely affected clutch of MZ*par2a* mutants immediately after natural spawning. After 1.5 h, all untreated embryos showed a lack of activation, with negligible chorion elevation and no overt blastodisc (Fig 4C). However, individuals from the same clutch treated with ionomycin showed strong significant rescue of egg activation, including significant chorion elevation, prominent blastodisc, and some blastomere cell division (Fig 4D and 4E). However, these cel divisions were abnormal, and treated *par2a* mutant embryos failed to successfully gastrulate. Treatment of WT with this concentration of ionomycin also resulted in blastomeres of unequal sizes (Fig 4A, 4B, and 4E), suggesting that Ca$^{++}$ levels must be tuned appropriately for successful cleavage. A similar disruption of cleavage furrows by calcium ionophores has been noted in *Drosophila* spermatocytes [42].

As outlined above, IP$_3$ is sufficient to induce egg activation in a number of species including zebrafish [2,43]. IP$_3$ sponges or IP$_3$R inhibitors have demonstrated that IP$_3$ signaling is also necessary for egg activation in sea urchins, *Xenopus*, and mammals [12,44–46]. Lee, Webb [25] have shown that zebrafish blastomere cleavage is sensitive to the IP$_3$R inhibitors 2-aminoethyl diphenyl borinate (2-APB) and Heparin. We tested if earlier treatment of WT eggs derived from natural spawning to increasing doses of the IP$_3$R antagonist, 2-APB can also disrupt egg activation. At lower concentrations, embryos successfully raised chorions and formed a blastodisc, however, as previously shown, they showed disruption of cell division. At 100 μM, embryos failed to form a blastodisc, although they did show normal chorion elevation. Treatment with 500 μM 2-APB abolished egg activation entirely, with no blastodisc, and no chorion elevation (S6A–S6E Fig). Similar effects on zebrafish eggs have been seen upon injection of the IP$_3$R inhibitor IRBIT (*ahcyl1*) [47]. Thus, blastomere division and egg activation respond in a dose-sensitive manner to IP$_3$R inhibition and, as in other animal classes, IP$_3$ is both necessary and sufficient for egg activation.

### Par2a regulates Ca$^{++}$ during egg activation via IP$_3$

This led us to determine if Par2a was regulating Ca$^{++}$ in the egg via IP$_3$. Firstly, we tested the sensitivity of mildly affected *par2a* mutants to low levels of 2-APB, reasoning that mildly reducing IP$_3$R activity would only synergise with mild *par2a* mutants if the latter was also disrupting IP$_3$ signaling. Mild *par2a* mutants partially raised a chorion and had a limited

PLOS Biology

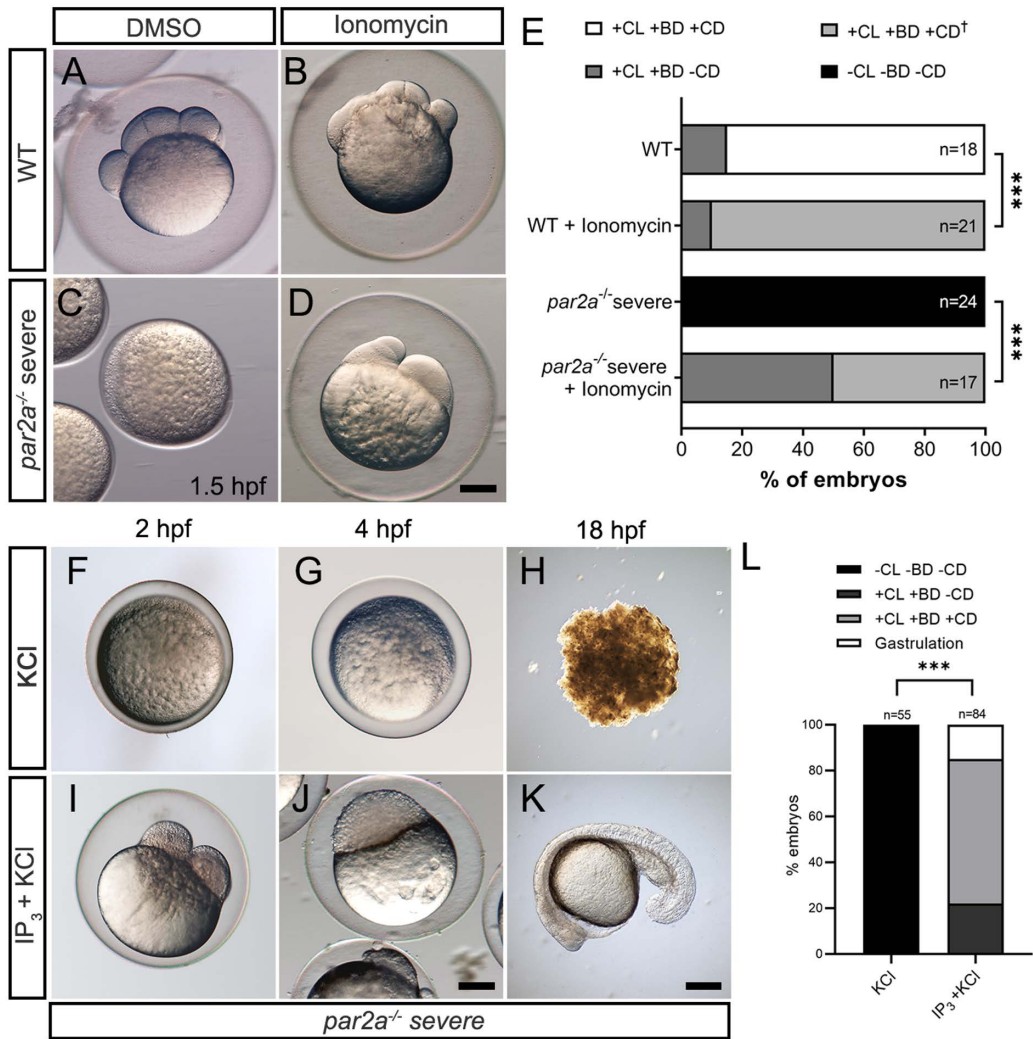

**Fig 4. Increasing intracellular Ca²⁺ either by Ionomycin or IP₃ rescues egg activation defects in *par2a* mutants. A–D:** Images of WT (A, B) or severe *par2a* mutants (C, D) embryos derived from natural spawning and treated with 0.1% DMSO (A, C) or 2 µM Ionomycin (B, D). **E:** Proportion of WT and severe *par2a⁻/⁻* embryos showing egg activation and blastomere division phenotypes at 1.5 hpf following treatment with 0.1% DMSO or 2 µM Ionomycin. Key: CL: Chorion Lift, BD: Blastodisc, CD: Cell Division, +: Present, −: absent. †: Abnormal. Chi-squared analyses; *** = $p < 0.001$. **F–K:** Images of naturally fertilized severe *par2a* mutant embryos injected with KCl alone (F, G, H) or with IP₃ in KCl (I, J, K) at 2 hpf (F, I), 4 hpf (G, J) and 18 hpf (H, K). **L:** Proportion of egg activation and blastomere division phenotypes in fertilized severe *par2a* mutant eggs injected with IP₃ in KCl or KCl alone. Key: CL: Chorion Lift, BD: Blastodisc, CD: Cell Division, +: Present, −: absent. Chi-squared analyses; *** = $p < 0.001$. Scale bars: D, J, K = 200 µm. See file S1 Data for underlying data. IP₃ is necessary for zebrafish egg activation.

blastodisc but aborted blastomere cytokinesis (S6H and S6J Fig). However, treatment with 50 µM 2-APB significantly enhanced the severity of the egg activation phenotype of mildly affected *par2a* mutants, with all embryos completely lacking a blastodisc and failing to raise a chorion at all, and in contrast to the mild effects of activation this concentration of 2-APB had on WT embryos (S6F, S6G, S6I, and S6J Fig). Thus, *par2a* mutants are acutely sensitive to mild reduction of IP₃R activity.

We also tested if supplementation of exogenous IP₃ ligand can rescue *par2a* defects as observed for serine protease inhibitors. Injection of 20 pmol IP₃ into naturally fertilized clutches of both mildly and severely affected *par2a* mutants was

able to rescue egg activation defects with high efficiency. In clutches derived from severe *par2a* females, injection of $IP_3$ led to all embryos raising a chorion and forming a blastodisc, whereas this never occurred in KCl-injected siblings (Fig 4F, 4I, and 4L). Many of these rescued embryos were able to undergo normal blastomere division and remarkably many could gastrulate to form a body axis at 18 hpf whereas all uninjected siblings failed to even generate a blastodisc and underwent necrosis by 18 hpf (Fig 4G, 4H, 4J, and 4L). Rescue of mildly affected *par2a* mutant clutches was even more pronounced, with almost 40% of embryos completing gastrulation, forming a normal body axis, and surviving past 24 hpf compared to KCl injected controls (S7A–S7C Fig). We conclude that *par2a* regulates $IP_3$ levels to ensure effective egg activation.

Par2 has been shown to regulate intracellular $IP_3$ and $Ca^{++}$ through coupling to G-protein alpha subunit Gq family members and activation of PLCβ [48,49]. By RNA-seq analysis, we determined that maternal transcripts for *gna11a* and *gna11b* are present at high levels in early zygotes, while paralogues of *gna14, gna15*, and *gnaq* are present at significantly lower levels. The most abundant PLC isozyme transcripts include *plcb3* and *plcb4* (S8 Fig and S2 Data). To test if these are also required for proper egg activation, we inhibited G alpha-q/11 protein function with YM-254890 [50] and PLC β with U-73122 [51] by incubation before and after egg activation. U-73122 potently aborted blastodisc formation and reduced chorion lifting. The effects of YM-254890 were more variable with some eggs showing reduced chorion elevation and blastodisc formation, while the majority generated a malformed blastodisc (S9A–S9D Fig and S3 Table).

We assessed if the differential expressivity of eggs derived from different *par2a^{lkc4}* mutant mothers was due to variations in expression of known components of $Ca^{++}$ regulation in oocytes, including heterotrimeric G proteins and PLC isozymes. We performed high throughput RNA sequencing of severe, mild, and sibling *par2a* zygotes, collected 1 h after natural crosses of mutant mothers to sibling fathers. As the genome only becomes transcriptionally active at mid-blastula transition, the only RNA present in early zygotes is that deposited during oogenesis. We included transcripts of SERCA, PMCA, $Ca_v3.2$, heterotrimeric G-alpha, $IP_3R$, SOCE, PLC, ryanodine receptors, and Trpm7 protein families but found few convincing differences in $Ca^{++}$ regulating genes between the severe and mild mutants, (S8 Fig and S2 Data). There was an approximately 4.4-fold and 3.3-fold increase in transcripts of *orai1b* (a component of SOCE) and the *plcxd1* PLC isozyme paralogue encoded by the *zgc:112023* transcript in the mild mutants compared to severe mutants (S8 Fig and S2 Data). Our RNA-seq data also confirmed the decay of mutant *f2rl1.1* transcripts in the *par2a* mutants as associated sequence fragments were reduced by over 80% in both mild and severe mutant zygotes (S10 Fig).

### Supplementing $IP_3$ restores cortical granule exocytosis and calcium dynamics in *par2a* mutants

Provision of $IP_3$ rescued chorion elevation in severe *par2a* mutants, hence we checked if this was due to restoration of CGE by examining cortical F-actin distribution and CG content using Phalloidin and FITC-MPL stain, respectively. KCl-injected severe *par2a* mutant embryos showed clusters of circular cortical F-actin staining which corresponded with high numbers of non-exocytosed CGs located at the cortex (Fig 5A). By contrast, $IP_3$-injected mutants displayed smooth and even cortical F-actin distribution with a significantly reduced number of CGs indicating the occurrence of CGE (Fig 5B and 5C). To confirm if exogenous $IP_3$ restores intracellular $Ca^{++}$ in *par2a* mutants, we compared in vivo $Ca^{++}$ dynamics using the T*g(actb2:GCaMP6s)^{lkc2}* reporter line in $IP_3$ and KCl injected severe *par2a* mutants. While KCl-injected controls showed only a small brief increase in $Ca^{++}$ transient, $IP_3$-injected *par2a* mutant embryos showed a significant spatial increase in $Ca^{++}$ levels with a longer wave duration (Fig 5D–5F and S10 Movie).

### Discussion

We have identified the first cell surface receptor directly required for $Ca^{++}$ regulation during egg activation. Our *par2a* maternal zygotic CRISPR mutants showed a strong egg activation defect leading to failure to raise a chorion and form a blastodisc. This was also evident in non-fertilized eggs, indicating this defect was not a fertilization block, rather disrupting egg activation specifically.

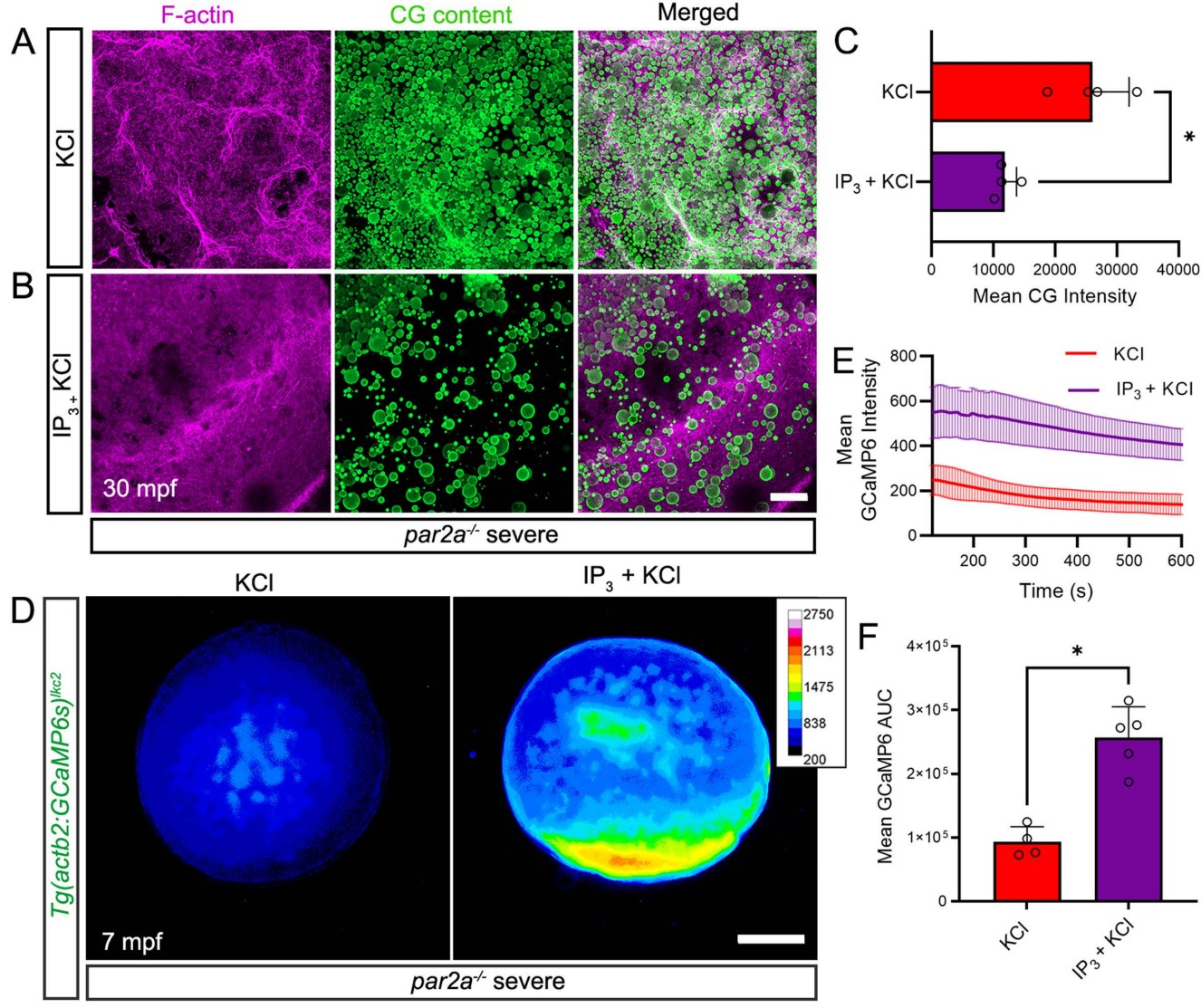

**Fig 5. IP$_3$ rescues egg activation defects by increasing intracellular calcium spatially. A, B:** Confocal images of F-actin (left; magenta), and cortical granule (middle; CG Content; green) distribution at the cortex of naturally fertilized severe *par2a* mutant eggs at 30 mpf. Eggs were injected with KCl alone (A) or with 20 pmol IP$_3$ in KCl (B). **C:** Quantification of CG content staining intensity in fertilized severe *par2a* mutant eggs injected with KCl alone (red bar) or with IP$_3$ in KCl (purple bar) $n = 4$; Mann–Whitney test; * = $p < 0.05$ **D:** Normalized pseudo-coloured fluorescent images of *Tg(actb2:G-CaMP6s)$^{lkc2}$* indicating calcium levels of naturally fertilized severe *par2a* mutant eggs at 7 min following injection with KCl alone (left) or with IP$_3$ in KCl (right). **E:** Changes in GCaMP6s intensity reported by the *Tg(actb2:GCaMP6s)$^{lkc2}$* transgene over 10 min in fertilized severe *par2a* mutants eggs following injection with KCl alone (red line) or with IP$_3$ in KCl (purple line). **F:** Graph of corresponding Area Under Curve analysis of GCaMP6s levels presented in (E). $n = 4, 5$; Mann–Whitney test; * = $p < 0.05$. Scale bars B = 50 μm; D = 200 μm. See file S1 Data for underlying data.

As a GPCR activated by extracellular serine protease activity, implication of a Par2 paralogue in egg activation unifies a number of disparate observations. Firstly, trypsin inhibitors block egg activation in fish species, including, as demonstrated here, in zebrafish [10]. We show that this is due to a significant and sustained reduction in Ca$^{++}$ waves in the egg. Complimenting this, addition of exogenous trypsin to Xenopus oocytes was previously demonstrated to liberate Ca$^{++}$ from internal stores [52]. It was shown subsequently that this activity is mediated by the xenopus G alpha-q and G alpha-14 homologs [53]. G alpha-q is also required for maximal Ca$^{++}$ release in the sea urchin [54]. Par2 has been implicated in activation of

G alpha-q/11 in both mouse and zebrafish [34–36], and experiments in xenopus did indeed propose that the effects of trypsin on egg Ca$^{++}$ were mediated by a trypsin receptor, even suggesting PAR-2 as the potential receptor [55]. Our injection experiments indicate that supplementation of IP$_3$ rescues both trypsin inhibition and loss of maternal Par2a, implying that a PLC isozyme lies downstream.

So far, two PLC isozymes have been implicated in egg activation. PLCζ is transferred directly into the egg from the sperm upon fertilization in mammals, while PLCγ is important in egg activation of non-mammalian species and is activated by cytosolic kinases such as those of the FAK and Src families [7]. The external stimulus activating these kinases is unclear. As G alpha-q/11 is a known activator of PLCβ, it is tempting to implicate this isozyme downstream of Par2a in the generation of IP$_3$, and thus Ca$^{++}$ release, during zebrafish egg activation. We identified that transcripts for *plcb3*, *plcd1b*, and *plcg1* are the most predominantly expressed PLC family members in the early zygote, and that broad PLC inhibition stalls egg activation. Defining the precise roles for each of the PLC enzymes in egg activation will likely require genetic approaches. Egg-derived PLCβ has been shown to be required for maximum amplitude of Ca$^{++}$ transients in mouse eggs [56]. This is similar to our observations of Par2a mutants, which show a reduced intensity of Ca$^{++}$ waves. In all movies, even in strong mutants, we noted the presence of a muted and brief calcium wave, which failed to propagate effectively. This would suggest that PLC activity may have a specific role in Ca$^{++}$ wave amplification and propagation in zebrafish, rather than initiation. Our RNA-seq analysis of zygotes demonstrated that G alpha-11 paralogues predominate over other G alpha-q family members. In addition, it also demonstrated that other G alpha proteins are expressed including G-protein alpha subunit, group I members, and G alpha-13 paralogues. While experiments in xenopus have also indicated that PLC activation might occur through a non-canonical mechanism, independent of heterotrimeric G-proteins [46,57], we show that egg activation is variably sensitive to a G alpha-q/11 family inhibition. The precise activation and roles of PLC enzymes in egg activation in fish and amphibia remain to be resolved.

As egg activation can occur without fertilization and requires Par2a, then the protease activating Par2a must be expressed in the egg as well. It is critical to identify this protease. Matriptase is a well-described protease activating Par2 [58,59], and our RNA-seq data showed that the genes encoding the Matriptase1 paralogues in zebrafish (*st14a* and *st14b*) are both present in zygotes (S8 Fig and S2 Data). We do not consider Matriptase1a or Matriptase1b to be relevant in activating Par2a in the egg as we have previously made both MZ*st14a* single and MZ*st14a;st14b* double maternal zygotic mutants and these showed only an ear otolith phenotype [35]. Aside from the identity of the protease, it is essential to determine how it is only activated upon spawning. One possible mechanism is simply a dilution away of serine protease inhibitors, known to be expressed in fish ovaries [9]. Similarly, release into water could alter the pH or ionic conditions that activate the protease. Alternatively, the protease could be sequestered and only released upon initial activation, where it can then cleave the extracellular domain of Par2a.

We found that the same *par2a* mutation had vastly different expressivity in eggs from different females. While there is little variation within a clutch, or between clutches from the same female, we saw that some females consistently gave MZ*par2a* clutches that were able to form a blastodisc and undergo limited, abnormal, blastomere cleavage. This allowed us to identify a later function for Par2a in regulating calcium dynamics at the blastomere cleavage furrow. No MZ*par2a* mutant ever gastrulated successfully to generate a normal axis. The reason for this variable expressivity is unknown. We hypothesized that compensation from maternal *par2b* might account for this variation, however some MZ*par2a; par2b* double mutant females also gave mildly affected clutches. This is not surprising as the genes are closely linked on Chromosome 21 and thus expressivity would be a trait linked to the original *par2a* mutation (and thus not appearing to sort independently). Additionally, our RNA-seq data indicated that *par2b* was expressed at about 12-fold lower levels than *par2a* in zygotes (S8 Fig and S2 Data), making it less likely to be contributing to egg activation. Comparative RNA-seq between mild and severe mutant zygotes failed to identify a clear basis for the expressivity at a transcriptional level. There did not appear to be compensatory upregulation of *par2b* or any other gene encoding a Protease-activated receptor, nor was there clear changes in known Ca$^{++}$ regulators aside from some upregulation of *orai1b* and a *plcxd1* paralogue. As the overall levels of these transcripts

are low it is not currently clear if these mild changes can account for the differential expressivity. Mildly affected eggs might have upregulated cryptic $IP_3$ or $Ca^{++}$ regulating pathways, such as undefined regulators of a PLC enzyme or even $IP_3$ receptors, and there may be compensation happening posttranscriptionally. PLCγ is regulated by phosphorylation and Src family kinases have been linked to calcium transients in zebrafish and xenopus oocytes [30,31]. It may be that in mild *par2a* mutants, there is an enhancement of the activity of a kinase activating PLCγ, through a mechanism yet to be defined.

At the blastomere cleavage furrow, $Ca^{++}$ is released in distinct waves corresponding to furrow positioning, propagation, and deepening [23]. Its precise functions are not fully understood although there is strong evidence it is required for cytoskeleton reorganization and vesicle exocytosis during apposition [23]. Analysis of the *nebel* mutant which has defective slow calcium waves has suggested that $Ca^{++}$ acts to propagate and deepen furrows through a Calmodulin and CamK mechanism [26]. The regulation of calcium at the furrow is unclear, with both ER and mitochondria sources available [26]. Previous work had indicated that $IP_3$ is essential for $Ca^{++}$ release during blastomere cleavage furrow deepening, and $IP_3$ receptors localize specifically to the furrow [25]. This explains why our ionophore treatments did not rescue *par2a* cell cleavages while $IP_3$ injection did. The ionophore would elevate $Ca^{++}$ globally while $IP_3$ would still provide normal restricted $IP_3R$ activation only at the furrow. SOCE components have also been shown to be present at the furrow and are required for sustaining calcium transients during the completion of furrow deepening and apposition during blastomere cytokinesis. [24,60]. Similarly, in mild *MZpar2a* mutants, $Ca^{++}$ transients at the incipient furrows are not sustained or propagated, leading to aberrant cytokinesis and mis-sized daughter cells. This identifies Par2a, along with SOCE, as an essential regulator of $Ca^{++}$ levels at the zebrafish blastomere cleavage furrow. In mammalian cells, the localization of PLCβ1 at the cleavage furrow also suggests a mechanism through G alpha-q/11 to regulate $IP_3$ levels [61], although there is no definitive identification of Par2 orthologues in mammalian blastocysts or oocytes.

Our discovery that a protease-sensitive cell surface receptor regulates calcium waves in the zebrafish egg provides a novel mechanism regulating egg activation and further demonstrates the diversity of approaches employed during fertilization across metazoa. While the presence of calcium waves is invariant in egg activation, the mechanisms initiating them are likely to vary with the diversity of egg activation environments. Protease activity is sensitive to the surrounding ionic conditions and thus might act as a sensor for egg release from the ovary into the spawning medium. Par2 is thus an effective link between protease activity and initiation of calcium waves. More generally, to what extent the role of Par2 in egg activation, and early embryo cell cleavage is conserved in other species and mammals is not yet clear. There may be significant mechanistic diversity even between fish species. For example, medaka eggs are activated only upon entry of sperm into the micropyle, and not simply upon release into water [62]. Hence activation and fertilization of medaka eggs are coupled more directly than in zebrafish, although Par2 could still play a role in $Ca^{++}$ wave propagation from sperm entry. In mouse, immunostaining hinted that Par2 is expressed on oocytes and can be activated by the sperm-derived protease, Acrosin [63], although there is no evidence from Par2 knockout studies that it is necessary for mouse egg activation. Prior to mammalian blastocyst implantation, the endometrium senses the presence of the invading embryo and initiates a number of intracellular signals, including waves of $Ca^{++}$. This has recently been shown to be mediated by endometrial expression of PAR2, which is activated by trypsin derived from the invading blastocyst, and which acts through PLC and $IP_3R$, as seen in zebrafish egg activation [64]. Whether PAR2 or related receptors also play a role in mammalian oocyte maturation itself or in blastocyte cell division, as seen in zebrafish, requires testing.

## Materials and methods

### Ethics statement

All animal experiments in this study were approved by the NTU Institutional Animal Care and Use Committee (IACUC) (IACUC number #A18002), conducted according to the guidelines of the Singapore National Advisory Committee for Laboratory Animal Research.

## Zebrafish maintenance and lines

Zebrafish lines were raised and maintained in NTU zebrafish facility under standard protocols (28 °C; 14 h-light/10 h-dark cycle) in compliance with guidelines provided by National Advisory Committee for Laboratory Animal Research. In-vivo Ca$^{++}$ imaging was performed using the *Tg(actb2:GCaMP6s, myl7:mCherry)*[lkc2] transgenic line (termed *Tg(actb2:G-CaMP6s)*[lkc2] hereafter) [35,38].

## Egg collection

Embryos collected through natural mating were incubated at 28 °C in E3 embryo medium (5 mM NaCl, 0.17 mM KCl, 0.33 mM CaCl$_2$, 0.33 mM MgSO$_4$). Un-activated/un-fertilized eggs were collected by squeezing eggs from female fish directly as per methods adapted from [65] following anesthesia with 0.02% Tricaine in E3 (MS-222 (Sigma-Aldrich) buffered to pH 7.0).

## In-vitro egg activation and fertilization

For experiments that did not require fertilization, eggs were covered in buffered Hank's saline (0.137 M NaCl, 5.4 mM KCl, 0.25 mM Na$_2$HPO$_4$, 1.3 mM CaCl$_2$, 1.0 mM MgSO$_4$, 4.2 mM NaHCO$_3$) containing 0.5% Bovine Serum Albumin (BSA) to prevent egg activation [37]. To initiate egg activation, 0.5% BSA was diluted out with 0.5× E2 medium (7.5 mM NaCl, 0.25 mM KCl, 0.5 mM MgSO$_4$, 75 μM KH$_2$PO$_4$, 25 μM Na$_2$HPO$_4$, 0.5 M CaCl$_2$, 0.35 mM NaHCO$_3$; buffered to pH 7.0 with HCl). For in vitro fertilization, dissected male testes were macerated in ice-cold buffered Hank's saline with 0.5% BSA. Eggs were fertilized in vitro by adding sperm suspension mixture to eggs, and then released for development by transferring to 0.5× E2 embryo medium. Staging of embryos were as per [66] while embryos prior to 1-cell stage were charted according to time from activation/fertilization.

## Generation of *par2* knockouts

Mutations in the *par2a* (also known as *f2rl1.1*; Gene ID: 794649) and *par2b* (also known as *f2rl1.2*; Gene ID: 100073342) genes were generated by CRISPR/Cas9 mutagenesis with single guide RNA (sgRNA) targeting exon 2 of each paralogue at a site corresponding with the second extracellular loop of the transmembrane helix determined by protein topology prediction provided by TMHMM v2.0 server [67]. Oligo templates for sgRNA transcription containing T7 promoter binding site was designed according to [68] utilizing the following crRNA and PAM (uppercase) sequence (*par2a* - 5′ gggtcgagt-gacatcatggcAGG 3′; *par2b* - 5′ ggagccctattattacttcaTGG 3′). The completed sgRNA template was obtained through PCR using TaKaRa PrimeSTAR Max DNA polymerase and transcribed with MEGAshortscript T7 Kit from Invitrogen. *Cas9* RNA was transcribed from *Not*I linearized pCS2-nCas9n plasmid [69] using the mMESSAGE mMACHINE SP6 Transcription Kit (Invitrogen). Following transcription, sgRNAs, and Cas9 RNA were treated with DNaseI and purified by precipitation with Sodium Acetate and Ethanol or Lithium Chloride. sgRNA and Cas9 RNA were diluted in 1× Danieau's buffer (5 mM HEPES (pH 7.6), 58 mM NaCl, 700 μM KCl, 400 μM MgSO$_4$.7H$_2$O, 600 μM Ca(NO$_3$)$_2$) with Phenol Red and injected into embryos at the 1-cell stage. A sample of injected embryos were sequenced at 24 hpf to evaluate mutagenesis efficacy and remaining embryos were raised to adulthood. To identify founders, adults were crossed and progeny sequenced for mutations. Individual alleles were incrossed to homozygosity. Double mutants (*par2a,b*) were generated by injecting *par2a* sgRNA into the established *par2b* mutant line. Genotyping primers and methods are given in S4 Table.

## RNA isolation, reverse transcription, and quantitative PCR

RNA was isolated from 1-cell zygotes derived from *par2a*[lkc4] homozygous females by extraction in TRIzol Reagent (Invitrogen) and precipitation in isopropanol. Complimentary DNA was generated using Superscript III Reverse Transcriptase (Invitrogen) with Oligo(dT)$_{20}$ primer. Quantitative PCR was performed on an Applied Biosystems StepOnePlus Real-Time

PCR System using KAPA SYBR FAST qPCR Master Mix (2×) (Kapa Biosystems) with ROX High Reference Dye (50×). qPCR was performed using primers for *par2a* and *eef1a1l1* transcripts (S5 Table) using 3 biological replicates and 3 technical replicates each.

## Bulk RNA-seq library generation and sequencing and analysis

*par2a^lkc4^* homozygous females were identified as generating severe or mildly affected embryos. These were crossed individually to sibling males to generate batches of severely affected and mildly affected *par2a* mutants and total RNA was isolated from zygotes as above. Total RNA was also extracted from zygotes of sibling females. mRNA was isolated from 1 µg total RNA using Oligo(dT) beads, and then fragmented using divalent cations and heat. Double-stranded cDNA was generated by synthesis with random primers. Following dA tailing, T-A ligation was used to add adaptors, and size selection performed using DNA Clean Beads. PCR was used to amplify samples and the libraries with different indices were multiplexed and loaded onto an Illumina NovaSeq6000 (S4 flow cell) for sequencing using a 2× 150 paired-end configuration as per manufacturer instructions.

Sequences were cleaned of primers, adaptors, and low-quality calls using Cutadapt (V1.9.1, phred cutoff: 20, error rate: 0.1, adapter overlap: 1 bp, min. length: 75, proportion of N: 0.1) and aligned to the zebrafish reference genome (assembly GRCz11) with Hisat2 (v2.2.1). HTSeq (v0.6.1) estimated gene expression levels from the pair-end clean data, while for differential expression analysis, the DESeq2 Bioconductor package was used. Fold change greater than 2 was considered with padj of genes set at ≤0.05 to detect differential expression. Heatmap was generated using ComplexHeatmap, a Bioconductor package, using R (ver. 4.4.2). Raw sequencing data is deposited in NCBI's Gene Expression Omnibus, GEO Series accession number GSE289416 (https://www.ncbi.nlm.nih.gov/geo/query/acc.cgi?acc=GSE289416).

## Generation of Par2a-eGFP transgenic line

The *egfp* coding sequence was cloned downstream of *par2a* to generate a Par2a-eGFP fusion construct which was then placed under the control of the 3.8kb *zp3.2* oocyte promoter [70] through use of the Tol2Kit Gateway cloning system [71]. Plasmid was injected into WT fish along with *Tol2* RNA to generate the transgenic line *Tg(zp3:par2a-egfp)^lkc8^*. Transgenic embryos of the founder female, generated by natural fertilization, were then imaged by Light-sheet microscopy.

## Fluorescent labeling

In vitro or naturally fertilized eggs were fixed at respective time points in 4% paraformaldehyde in PBS at 4 °C overnight. For visualizing CGs, fixed embryos were washed in PBST (0.1% Tween 20 in PBS), permeabilized in 0.2% Triton-X in PBS for 2 h, dechorionated and labeled by incubation overnight at 4 °C in PBST with 50 µg/ml FITC-MPL (F-3901-1, EY Laboratories). To examine cortical F-actin distribution, embryos were incubated overnight at 4 °C in PBST containing a 1:500 dilution of Alexa Fluor-546 Phalloidin (A12379, Invitrogen). Where possible, FITC-MPL and Phalloidin stainings were performed simultaneously. Embryos were subsequently washed in PBST and stored in glycerol for imaging.

## Tracking cytoplasmic streaming

Embryos collected from natural spawning were injected at the vegetal pole at 5 mpf with 1 nl of 1:5 fluorescent yellow-green carboxylate-modified polystyrene latex beads (L4530, Sigma-Aldrich) diluted in $H_2O$. Embryos were then immediately mounted in 0.8% Low Melt Agarose (LMA) (Mo Bio Laboratories) in 0.5× E2 medium and orientated laterally on 15 mm glass-bottom imaging dishes (NEST Biotechnology, 801002) for imaging. Distance of travel was measured by taking two images on an upright Zeiss AxioImager M2 microscope at 0 and 60 mins after mounting. Time-lapse movies of bead movement were taken using an upright Zeiss LSM800 confocal microscope with a 45 s interval time.

## Inhibitor and rescue treatments

For inhibitor treatments of eggs, squeezed eggs were held in Hank's saline containing 0.5% BSA, sperm solution added, and subsequently released for egg activation and fertilization by washing with 0.5× E2 embryo medium containing inhibitor solution. To inhibit $IP_3R$, 2-APB (D9754, Sigma-Aldrich) was dissolved in Methanol and used at 50, 100, or 500 μM. 0.2% Methanol was used as control. To inhibit Gq/11, eggs were squeezed from WT female and held in Hank's saline containing 0.5% BSA. This solution was then replaced with Hank's saline +0.5% BSA containing 200 μM YM-254890 (10-1590-0100, Focus Biomolecules, dissolved in DMSO), and the eggs incubated for 20 min. Activation was then conducted in E2 Medium with 200 μM YM-254890). To inhibit PLCβ, eggs were treated as above but replacing YM-254890 with 100 μM U-73122 (70740, Cayman Chemicals, dissolved in DMSO).

For protease inhibitor treatment, Aprotinin (14716, Cayman Chemical) and Benzamidine Hydrochloride Monohydrate (Benzamidine HCl) (B-050, Gold Biotechnology) were dissolved in $H_2O$. and diluted in 0.5× E2 embryo medium before adding to suspended egg sperm mix. Aprotinin and Benzamidine HCl were used at 5 mg/ml and 44 mg/ml, respectively.

For Ionomycin treatment, naturally fertilized eggs were incubated in 0.5× E2 supplemented with 2 μM Ionomycin (I0634, Sigma-Aldrich; dissolved in DMSO) with 0.1% DMSO as control. For $IP_3$ rescue, 2 nl of a 10 mM $IP_3$ solution (D-myo-Inositol 1,4,5-tris-phosphate trisodium salt; I9766, Sigma-Aldrich; dissolved in 0.7 mM KCl) was injected into the yolk of naturally fertilized eggs. Injected eggs were observed at different time points as stated. For control comparison, eggs were injected with 0.7 mM KCl solution only.

## Microscopy

Images were taken on an upright Zeiss AxioImager M2 compound microscope, an upright Zeiss LSM800 confocal microscope, a Zeiss AxioZoom V16 microscope and a Zeiss Light-sheet Z.1 microscope. Images were processed on Zen (ver. 2.3 Lite) or using Fiji (ImageJ, v1.53t). Embryos and eggs were mounted laterally in 0.8% LMA in 0.5× E2 medium in 15 mm glass-bottom dishes (Nest Scientific).

Live low power images of embryos in E2 medium were taken using a Zeiss AxioZoom V16 microscope. Nomarski imaging was performed on the Zeiss AxioImager M2 microscope.

For visualizing early calcium dynamics, 0.5× E2 activated unfertilized eggs squeezed from *Tg(actb2:GCaMP6s)*[lkc2] females or $IP_3$ injected eggs were imaged directly on a Zeiss AxioZoom V16 microscope without mounting. For visualizing Par2a-eGFP and calcium dynamics at the cleavage plane, *Tg(zp3:par2a-egfp)*[lkc8] and *Tg(actb2:GCaMP6s)*[lkc2] embryos, collected by natural spawning, were dechorionated manually and mounted in 1% LMA in a 50 μl volume capillary (BRAND 701908). Fluorescent time-lapse series was then captured on a Zeiss Light-sheet Z.1 microscope.

## Image processing and quantification of GCaMP6 fluorescence

Mean intensities of projected fluorescent images were analyzed using the Average Intensity function in Fiji. For fluorescent time-lapse images, an average intensity plot across time was generated by measuring the mean fluorescence normalized to background fluorescence, for each time frame. For each frame, a region of interest corresponding to the embryo outline was generated by a Create Selection function in Fiji. Images and timelapses of GCaMP6 fluorescence are presented pseudo-coloured using the "16 colors" Lookup Table from Fiji.

## Statistical analysis

Graphpad Prism software was used for statistical analysis. All graphs represent mean with standard deviation. Statistical tests used were Student $t$ test, Mann–Whitney test, Area Under Curve with Mann–Whitney test, and Chi-squared test, as stated in figure legends. *P*-values were presented as $* = p < 0.05$, $** = p < 0.01$, or $*** = p < 0.001$, with 0.05 set as alpha in all. Underlying data of all graphs are presented in file S1 Data.

## Supporting information

**S1 Fig. The serine protease inhibitor Benzamidine aborts egg activation and reduces Ca²⁺ wave propagation. A**: Eggs fertilized in vitro in E2 medium with or without 40 mg/ml Benzamidine HCl. **B**: Proportion of embryos showing egg activation and blastomere division phenotypes for 40 mg/ml Benzamidine HCl treatment vs. E2 alone. Key: CL: Chorion Lift, BD: Blastodisc, CD: Cell Division, +: Present, −: absent. Chi-squared analyses; *** = $p < 0.001$; $n = 100$. **C**: Normalized pseudo-coloured fluorescent images of unfertilized *Tg(actb2:GCamP6s)^lkc2* eggs indicating Ca²⁺ dynamics at 250 s during egg activation in E2 or Benzamidine HCl. **D**: Quantification of changes in mean GCamP6s intensity from fluorescent timelapse of *Tg(actb2:GCamP6s)^lkc2* eggs activated in E2 (gray line) vs 40 mg/ml Benzamidine HCl treatment (red line) **E**: Corresponding statistical analyses of GCamP6s intensity AUC from (D). $n = 4$; Mann–Whitney test; * = $p < 0.05$. Scale bars: A = 500 μm, C = 200 μm. See file S1 Data for underlying data.
(TIF)

**S2 Fig. Generation of *par2a* and *par2b* mutant alleles. A, B**: CRISPR mutagenesis of *par2a* (A) and *par2b* (B). Intron-Exon structures are given above. Boxes correspond to exons and dashed lines represent introns. Both genes have 2 exons. Dark blue and yellow boxes represent coding and untranslated regions respectively. Red bars indicate the approximate location of CRISPR target site with the sequence given below for each allele (name and mutation summary given on left), with corresponding WT sequence for comparison. Blue bars designate CRISPR binding sites and gray bars indicate PAM site. Red and green text indicate deleted and inserted nucleotides respectively. Brown amino acid sequences indicate novel amino acids introduced by frameshift. The 103 bp insertion in the *par2a^lkc4* allele shows homology to the Cas9 plasmid template used to synthesize Cas9 RNA. **C**: Protein schematics for all Par2a and Par2b CRISPR alleles. Green, red, blue, purple, and orange sections indicate signal peptide, tethered inhibition domain, 7-pass transmembrane region, intracellular tail, and induced frameshift regions, respectively. Corresponding allele names and summary given below.
(TIF)

**S3 Fig. Reduction of *par2a* RNA in mutant zygotes.** Relative levels of *par2a* mRNA in 1-cell zygotes derived from *par2a^lkc4/lkc4* homozygous mothers and sibling mothers. Quantitative PCR was performed on cDNA, and *par2a* transcript levels normalized to *eef1a1l1* transcripts. RNA was prepared from pooled zygotes derived from three *par2a^lkc4/lkc4* homozygous mothers and three sibling mothers. All were outcrossed to wild-type fathers. $n = 3$; unpaired two-tailed *t* test; *** = $p < 0.001$. See file S1 Data for underlying data.
(TIF)

**S4 Fig. Aborted egg activation and cell division in *par2a* mutants.** Nomarski images of naturally fertilized (**A, C, E**) and unfertilized (**B, D, F**) eggs derived from WT (A, B), mild (C, D), and severe (E, F) *par2a* mutant females. Eggs were fertilized with WT sperm in A, C, E. Images were taken at 5 min (5 mpa), 30 min (30 mpa), 1 h (1 hpa), and 1.5 h postactivation (1.5 hpa). Chorion elevation is reduced in the *par2a* mutant derived eggs (C–F) and a defective blastodisc forms only in the mild *par2a* mutant eggs (C, D), with none forming in either the fertilized or unfertilized severe mutant eggs (E, F). Scale bar: F = 200 μm.
(TIF)

**S5 Fig. Loss of *par2a* generates egg activation defects. A–C**: Nomarski images of WT (left) and mild *par2a⁻/⁻* mutants (right) derived from natural crosses at 2 hpf (A) and sphere stage (4 hpf; B). **C**: DAPI staining superimposed on Nomarski images of mild *par2a⁻/⁻* mutants at sphere stage **D**: Clutches of embryos from WT, mild and severe *par2a⁻/⁻* mutants generated by IVF and imaged at 0.5, 5, 15, 30, and 60 mpf. **E**: Projected confocal images showing dynamics of cortical F-actin stained by AlexaFluor-546 Phalloidin at 0.5, 2, and 5 mpa, in E2 activated WT (top), *par2a⁻/⁻* mild (middle) and severe *par2a⁻/⁻* mutant eggs (bottom). **F**: Projected confocal images showing dynamics of Cortical Granule release stained by FITC-MPL at 0.5, 2, and 5 mpa, in E2 activated WT (top), and *par2a⁻/⁻* mild mutants (bottom) **G**: Distance

traveled by injected fluorescent latex beads in individual fertilized eggs of WT (top; green) vs. mild *par2a*⁻/⁻ mutants (bottom; yellow) from start point (left) to end position at 1 h. Each line represents the distance traveled by tracked beads in each embryo. Scale bars: C = 200 μm; D = 500 μm; E, F = 50 μm. See file S1 Data for underlying data.
(TIF)

**S6 Fig. Egg activation of *par2a* mutants is highly sensitive to IP₃R antagonism. A–D**: WT embryos treated with methanol carrier (A), or 50 μM (B), 100 μM (C), and 500 μM (D) of IP₃R inhibitor, 2-APB. **E**: Proportion of egg activation phenotypes presented by different concentrations of 2-APB. Key: CL: Chorion Lift, BD: Blastodisc, CD: Cell Division, +: Present, −: absent. Chi-squared analysis; *** = $p < 0.001$. **F–I**: Naturally fertilized WT (F, G) and mild *par2a* mutant (H, I) embryos treated with methanol (F, H) or a low dose (50 μM) of 2-APB. Low dose of 2-APB strongly exacerbates egg activation defects in mild *par2a* mutants. **J**: Counts of proportion of embryos in WT and mild *par2a* mutants showing extent of phenotype in low dose of 2-APB. Key: CL: Chorion Lift, BD: Blastodisc, CD: Cell Division, +: Present, −: absent, ½: half reduced; Chi-squared analysis; *** = $p < 0.001$. Scale bars: D, I = 500 μm. See file S1 Data for underlying data.
(TIF)

**S7 Fig. Injection of IP₃ rescues mild *par2a* mutants and restores body axis. A, B** Lateral Nomarski images of naturally fertilized 24 hpf mild *par2a* embryos injected with either KCl (A) or 20pmol IP₃ in KCl (B). **C**: Proportion of mild *par2a*⁻/⁻ embryos from a single clutch failing to gastrulate or showing full or partial body axis following KCl or IP₃ injection. Chi-squared analysis; *** = $p < 0.001$. Scale bar: B = 200 μm. See file S1 Data for underlying data.
(TIF)

**S8 Fig. Heatmap of calcium regulator gene expression in *par2a* mutant zygotes.** Heatmap of transcript levels of genes involved in calcium regulation in early zygote and blastula stages. Transcript levels were derived by high throughput RNA-seq of mRNA from severe and mild *par2a*^lkc4^ mutants and sibling zygotes. Genes are grouped by family in rows and triplicate biological replicates are grouped by columns. See file S2 Data for underlying data.
(TIF)

**S9 Fig.** Inhibition of G alpha-q/11 and PLCβ disrupts egg activation. **A–D** Eggs squeezed from WT females and held in Hank's buffered saline with BSA. Controls were then activated with E2 medium (A, C). To inhibit G alpha-q/11, eggs were incubated in 200 μM YM-254890 in Hank's buffer for 20 min then treated with E2 containing 200 μM YM-254890 (B). To inhibit PLCβ, eggs were incubated in 100 μM U-73122 in Hank's Buffer for 20 min then treated with 100 μM U-73122 in E2 (D). Scale bars: B, D = 500 μm.
(TIF)

**S10 Fig. Reduction in *f2rl1.1 (par2a)* transcript levels in *par2a* mutant zygotes.** FPKM counts for *f2rl1.1 (par2a)* transcripts from High throughput RNA sequencing of severe *par2a* (red circles), mild *par2a* (orange circles), and sibling zygotes. Each biological replicate shown as a single point. See file S2 Data for underlying data.
(TIF)

**S1 Table. Expressivity of mutant phenotypes in different sisters and brothers of *par2a* alleles.**
(DOCX)

**S2 Table. Consistent expressivity of phenotypes between different clutches from individual *par2a* mutant females.**
(DOCX)

**S3 Table. Egg activation phenotypes induced by Gq/11 and PLCβ inhibitors.**
(DOCX)

**S4 Table. Genotyping primers and methods.**
(DOCX)

**S5 Table. qPCR primers.**
(DOCX)

**S1 Data. Numerical values for all plotted data in the main and supporting figures.** Worksheet name corresponds to each figure and panel number.
(XLSX)

**S2 Data. Gene expression in sibling, severe *par2a* and mild *par2a* mutants at1-cell zygote stage.** Spreadsheets of total and differential gene expression levels measured by RNA-seq in sibling, severe *par2a* and mild *par2a* mutant zygotes (annotated in columns). All genotypes were analyzed in triplicate. Raw RNA sequence fragment counts and fragments per kilobase of exon per million reads mapped (FPKM) of genes (named by row) are given in alternating columns. Spreadsheet 1 (all.fpkm_anno) lists all genes. Spreadsheet 2 (Group_Sib-VS-StrMut_DE_signific). list only genes significantly differentially expressed between siblings and severe mutants. Spreadsheet 3 (Group_Sib-VS-WeakMut_DE_signifi). list only genes significantly differentially expressed between siblings and mild mutants. Spreadsheet 4 (Group_StrMut-VS-WeakMut_DE_sign). list only genes significantly differentially expressed between severe and mild mutants.
(XLSX)

**S1 Movie. Comparable rapid Ca$^{2+}$ waves occur following egg activation by fertilization or by medium alone.** Fluorescent timelapse of eggs from a *Tg(actb2:GCaMP6s)$^{lkc2}$* female activated by IVF in the presence of sperm (left panel) or activated parthenogenetically in sperm-free E2 only (right panel). The 70 s before activation and the 400 s following egg activation are shown. Scale bar = 100 μm.
(AVI)

**S2 Movie. Inhibition of serine protease activity by Aprotinin reduces Ca$^{2+}$ wave intensity and duration during egg activation.** Fluorescent timelapse of unfertilized eggs from a *Tg(actb2:GCaMP6s)$^{lkc2}$* female activated in E2 (left panel) or 5 mg/ml Aprotinin (right panel). First 620 s following egg activation are shown. Scale bar = 200 μm.
(AVI)

**S3 Movie. Inhibition of serine protease activity by Benzamidine reduces Ca$^{2+}$ wave intensity and duration during egg activation.** Fluorescent timelapse of unfertilized eggs from a *Tg(actb2:GCaMP6s)$^{lkc2}$* female activated in E2 (left panel) or 40 mg/ml Benzamidine (right panel). First 620 s following egg activation are shown. Scale bar = 200 μm.
(AVI)

**S4 Movie. *par2a* mutant eggs show defective egg activation.** Time-lapse movies of naturally fertilized (top row) and unfertilized activated (bottom row) eggs of WT (left), mild *par2a$^{-/--}$* (middle), and severe *par2a$^{-/--}$* (right) mutants over the first 120 min following fertilization/activation. Scale bar = 200 μm
(AVI)

**S5 Movie. *par2a* mutant embryos show failed cytoplasmic streaming.** Time-lapse movies of movement of fluorescent latex beads injected into the vegetal region of WT (left) and severe *par2a$^{-/--}$* mutants immediately following natural fertilization. Beads were tracked (red arrowheads) over 55 min. Scale bar = 100 μm
(AVI)

**S6 Movie. Mild *par2a* mutant eggs show reduced Ca$^{2+}$ wave intensity and duration during egg activation.** Fluorescent timelapse of GCaMP6s dynamics during WT (left) and mild *par2a* mutant (right) egg activation as visualized

by the *Tg(actb2:GCaMP6s)^lkc2* transgenic line. Eggs were activated in E2 and imaged for the first 695 s. Scale bar = 200 μm.
(AVI)

**S7 Movie. Severe *par2a* mutant eggs show strongly reduced Ca²⁺ wave intensity and duration during egg activation.** Fluorescent timelapse of GCaMP6s dynamics during WT (left) and severe *par2a* mutant (right) egg activation as visualized by the *Tg(actb2:GCaMP6s)^lkc2* transgenic line. Eggs were activated in E2 and imaged for the first 745 s. Scale bar = 200 μm.
(AVI)

**S8 Movie. Mild *par2a* mutant eggs show reduced Ca²⁺ wave intensity and propagation at the blastomere cleavage furrow.** Fluorescent timelapse of GCaMP6s dynamics during WT (left) and mild *par2a* mutant (right) blastomere cleavage as visualized by the *Tg(actb2:GCaMP6s)^lkc2* transgenic line. Eggs were naturally fertilized and imaged for 60 min. Scale bar = 200 μm.
(AVI)

**S9 Movie. Par2a-eGFP localizes to the cleavage furrow and cell membrane of dividing blastomeres.** Fluorescent timelapse of Par2a-eGFP dynamics during blastomere cleavage of the *Tg(zp3:par2a-egfp)^lkc8* transgenic line. Eggs were naturally fertilized by crossing to WT males and imaged from the animal pole for 120 min. Scale bar = 100 μm.
(AVI)

**S10 Movie. IP₃ can partially restore Ca⁺⁺ wave intensity and duration during egg activation in severe *par2a* mutant eggs.** Fluorescent timelapse of GCaMP6s dynamics during egg activation of severe *par2a* mutants as visualized by the *Tg(actb2:GCaMP6s)^lkc2* transgenic line. Eggs were injected with KCl alone (left) or IP₃ in KCl and imaged for the first 735 s following natural fertilization. Scale bar = 200 μm.
(AVI)

## Acknowledgments

We thank Bernett Lee for generating the RNA-seq Heatmap. We are grateful to the Nanyang Technological University Optical Bio-Imaging Centre (NOBIC) and the NTU Zebrafish Facility.

## Author contributions

**Conceptualization:** Tom J. Carney.

**Data curation:** Jiajia Ma.

**Formal analysis:** Jiajia Ma, Tom J. Carney.

**Funding acquisition:** Tom J. Carney.

**Investigation:** Jiajia Ma.

**Methodology:** Jiajia Ma.

**Supervision:** Tom J. Carney.

**Writing – original draft:** Jiajia Ma, Tom J. Carney.

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
