## [Editor Report · Decision Letter 0]

5 Jun 2024

Dear Dr Carney,

Thank you for submitting your manuscript entitled "Protease-activated receptor 2 links protease activity with calcium waves during egg activation and blastomere cleavage" for consideration as a Research Article by PLOS Biology.

Your manuscript has now been evaluated by the PLOS Biology editorial staff as well as by an academic editor with relevant expertise and I am writing to let you know that we would like to send your submission out for external peer review.

Once your full submission is complete, your paper will undergo a series of checks in preparation for peer review. After your manuscript has passed the checks it will be sent out for review. To provide the metadata for your submission, please Login to Editorial Manager (https://www.editorialmanager.com/pbiology) within two working days, i.e. by Jun 07 2024 11:59PM.

Kind regards,

Ines

--

Ines Alvarez-Garcia, PhD

Senior Editor

PLOS Biology

---

## [Decision Letter · Decision Letter 1]

15 Aug 2024

Dear Dr Carney,

Thank you for your patience while your manuscript "Protease-activated receptor 2 links protease activity with calcium waves during egg activation and blastomere cleavage" was peer-reviewed at PLOS Biology. It has now been evaluated by the PLOS Biology editors, an Academic Editor with relevant expertise, and by several independent reviewers.

In light of the reviews, which you will find at the end of this email, we would like to invite you to revise the work to thoroughly address the reviewers' reports.

As you will see below, the reviewers were generally positive about your study and thought that the data showing a role for Par2 in zebrafish egg activation is solid. However, they also have some concerns. Specifically, they had some questions about the genetics of the studied zebrafish, which we would like you to address with a better analysis of the genetics (Reviewer 2) or a par2 rescue experiment, as suggested by Reviewer 3. Furthermore, we would like you to provide some more mechanistic insight, as suggested by Reviewer 1. We suggest that you focus either on the receptor activation or on the signaling pathways downstream of the receptor. We would like to invite a revision that addresses the concerns of the reviewers. After discussion with the academic editor, we don't think that providing data on the timing of Ca2+ release (R3, point2), the location of the putative receptor (R3, point5), or the difference between ionophore exposure and IP3 (R3, point 4) are necessary for the publication of this manuscript.

Given the extent of revision needed, we cannot make a decision about publication until we have seen the revised manuscript and your response to the reviewers' comments. Your revised manuscript is likely to be sent for further evaluation by all or a subset of the reviewers.

**IMPORTANT - SUBMITTING YOUR REVISION**

*Re-submission Checklist*

*Published Peer Review*

*PLOS Data Policy*

*Blot and Gel Data Policy*

Sincerely,

Suzanne

Suzanne de Bruijn, PhD

Associate Editor, PLOS Biology

Sbruijn@plos.org

On behalf of,

Ines

Ines Alvarez-Garcia, PhD

Senior Editor

PLOS Biology

REVIEWS:

Reviewer #1: This paper tests the role of par2 receptors in egg activation in zebrafish. The authors show that par2 mutants have defects in egg activation that can be rescued by IP3 injection. The mutants convincingly show a role for par2 in egg activation, however, the mechanism of activation of the receptor are not clear and the signaling cascade downstream of the par2 receptor are not defined. That is which Galpha and/or PLC are involved. Some important controls and validations are needed such as measures of fertility of par2 mutant females and validation of expression levels of par2 mutant (and confirmation of no WT protein expression that is validation in the CRISPR mutants). It would also be important to test the expression levels of PLCs, Galphas, IP3 receptor isoforms, and other Ca2+ transporters in the par2 mutants as compared to WT as this may help define the differential phenotype in some par2 mutant females.

Specific comments:

A body of work describing the differentiation of Ca2+ signaling in preparation for fertilization in xenopus is quite relevant to the introduction and needs to be cited (Machaca, 2007; Nader et al., 2013)

Need to show Western blots from eggs confirming the generation of the mutants par2 proteins and their penetrance at the protein level in eggs.

Also some measure of the fertility rates of female par2-/- needs to be assessed to confirm that it correlates with the observed phenotypes at the egg level. Male fertility for these mutants also needs to be assessed to support the conclusion that the phenotype is egg specific. I am confused a little here as the authors state that par2 mutants never gastrulate which implies that homozygous par-/- females should not produce viable offspring? Yet they also state that par2 mutant animals reach adulthood normally? Please clarify.

The differential phenotype severity in par2 mutant females is attributed to 'differential expressivity'. As the females are homozygous for a non-functional par2 mutant this differential expressivity is likely due to other pathway partially compensating. Some discussion of what those could be would be useful as par2b is ruled out.

Fig. 4: treating embryos with ionomycin to increase intracellular Ca2+ as mentioned is non-specific and does not appropriately maintain the spatial and temporal dynamics of the physiological Ca2+ signal at the cleavage furrow. A more elegant and better controlled experiment, if technically doable, would be to inject one of the two blastomeres with IP3 and assess division as the authors are postulating a par2 depend Ca2+ release mechanism IP3 inject should have a more specific rescue effect.

Fig. S5: 2APB is not a specific IP3R inhibitor and blocks among other Ca2+ signaling proteins store-operated Ca2+ channels. As such it cannot be used to conclude that IP3R are involved. A more specific blocker such as xestospongin is needed.

Machaca, K. 2007. Ca2+ signaling differentiation during oocyte maturation. J. Cell. Physiol. 213:331-340.

Nader, N., R.P. Kulkarni, M. Dib, and K. Machaca. 2013. How to make a good egg!: The need for remodeling of oocyte Ca(2+) signaling to mediate the egg-to-embryo transition. Cell Calcium. 53:41-54.

Reviewer #2: Ma & Carney demonstrate a role for Par2 in zebrafish egg activation, presumably acting upstream of IP3 to regulate Ca++ release. Their work is a solid contribution to our understanding of Ca++ regulation in the egg and helps elucidate the mechanism underlying how protease activity triggers the signaling cascade of fertilization/activation.

The only moderate concern is regarding the variable expressivity of the defects in par2a-/- eggs. While in my opinion the authors do not need to conclusively explain this phenomenon in this manuscript, some additional reporting of the genetics would help contextualize this result. Specifically:

- How many independent par2a alleles were evaluated in par2a-/- eggs? The text reports three (second paragraph of "Maternal par2a mutants..." section), but there are only 2 reported in Supp Table 1, with the 3rd allele (lkc5) presumably generated in a par2b background, according to the methods.

- To what extent were multiple clutches from the same par2a-/- female evaluated for consistent degree of expressivity? This is reported in passing, but no data are shown (Supp Table 1 seems to report numbers for only a single clutch per female)

- What is the relationship between the par2a-/- females of the same allele? Are they siblings? Or were there multiple different parents?

- The methods suggest that F0 founders may have been incrossed to potentially yield homozygous (or rather, transheterozygote/heteroallelic) F1. Can the authors clarify the generations and the genotyping that yielded the final par2a-/- females whose eggs were evaluated?

- What primers were used for genotyping / sequencing?

- The large insertion in the lkc4 allele appears to be cas9 sequence, suggesting that cas9 DNA was available in the cell to serve as a repair template... The lkc7 par2b allele is also quite unusual given how far the lesion is from the PAM site. If there are any additional methodological details that should be documented, I encourage the authors to do so

Additional minor clarifications/edits:

- In the intro and results (last paragraph of "Maternal par2a mutants...") the authors state that egg activation in fish is triggered by spawning/water; however, this is not the case for all fish -- notably medaka eggs do not activate in water but rather coupled with fertilization (Iwamatsu 1991)

- par2a/b have updated gene symbols, f2rl1.1 and f2rl1.2 respectively, so the authors should report these as well as the relevant accession numbers (Refseq? Ensembl?) that they based their gene models on

- The authors may want to note that par2b seems to be only weakly maternal in constrast to par2a according to RNA-seq datasets (e.g., White et al 2017), which may account for its lack of involvement in egg activation.

- matriptase1 (st14a/b) is also maternal, perhaps relevant to the discussion in speculating what the maternal protease may be

- par2a and par2b are tandem duplicates very close to each other on the chromosome, which has implications for the genetics (i.e., mutating one locus may have a regulatory consequence on the other); but also an intriguing possibility for functional divergence due to duplication, given that Par2b's role in inflammation may be more conserved

- In the GCaMP fluorescence figures, please indicate in the legend that the images are pseudocolored and what the intensity values represent (normalized? ratio?)

- Fig 3D, these are animal view? Indicate this in the legend

- In Supp Fig 2a/b, the positions of the blue bars are off compared to the sequences of the gRNAs reported (they are too short and slightly shifted)

Reviewer #3: In this manuscript, the authors investigate the mechanism of egg activation in fish eggs. They use data available in the literature to inquire if proteases and/or their receptors could be involved in egg activation in this system. The most solid evidence in the literature to date is that protease inhibitors prevented fish egg activation and the progression of fertilization in fish of several species. Then, the authors examined and confirmed that this also happens in Zebrafish. Next, they explored the possible pathways involved. They reported that in the presence of Aprotinin or similar inhibitors, fertilization failed to induce egg activation, and embryo development failed to progress. Following evidence from somatic cells in the zebrafish epidermis, where Par2b produces IP3 following activation by a protease, the authors hypothesized that Par2a or b may also produce IP3 in fish eggs. These Par2 are G-protein coupled receptors and presumably can generate the IP3 necessary to trigger Ca2+ release. The authors generated Par-2mutants using CRISPR/Cas9 and found that the eggs of homozygous mutant females for Par-2a showed defects in egg activation following fertilization. The defects are like those observed in the presence of protease inhibitors, and fertilized mutant Par-2a embryos showed poor egg activation, few gastrulated, and none survived 5dpf. They found two phenotypes, a more severe one and a milder one, and the molecular differences between them remain unclear. The authors then examined other events of egg activation, such as Ca2+ release, cytoplasmic streaming, and cleavage, and all were defective in the mutants, especially in the severe form. Finally, it is shown that the defects in Ca2+ release were due to inhibition or lack of the IP3 signaling system. Ca2+ ionophore and exogenous IP3 rescued the activation phenotype, but only IP3 supported full embryo cleavage, and these embryos could gastrulate.

Overall, the data in this manuscript are solid, with a strong rationale and methodology. Some statements in the text need additional crafting. Additional information may be necessary to answer some critical remaining questions.

Main points:

1- The timing of the Ca2+ rises could be better presented. In all figures, the concentrations of Ca2+ seem to be rising from the outset. Could the authors inhibit fertilization with BSA and then wash it off while monitoring Ca2+? In this manner, the Ca2+ rise can follow an established baseline.

2- The authors could demonstrate whether the timing of Ca2+ release and egg activation are the same and whether induced by spawning or by spawning + fertilization. In other words, is the Ca2+ signal modified at all by the presence of fertilization?

3- Similarly, how the protease receptor is activated is not known. Is there a protease in the CGs? This is something they propose. The authors could block CG exocytosis or reduce it using pharmacological inhibitors (Myosin II inhibitors) and examine if there is any difference in egg activation. This would get close to answering whether the protease is a component of the cortical granules. If it is not affected, it could be that this receptor is constitutively active, and the simple dilution effect may activate it. Have the authors used different ratios of eggs to media to see if this affects the rate of activation and Ca2+ elevation?

4- The authors should address the issue of why the ionophore exposure rescues egg activation but not cleavage, whereas IP3 does both. Does IP3 persist from the initial bolus injection and support those initial cleavages? The failure of Ionomycin would suggest that the internal PLC is not sensitive to Ca2+.

5- What about the location and expression of the putative receptor? Is there an antibody to demonstrate expression and distribution?

6- Can the expression of the receptor into the Par mutants rescue the phenotype? This will be important and, equally, the receptor's expression levels could be titrated.

Minor points:

1. In the introduction, a better explanation of when fertilization occurs in the context of egg activation will be beneficial. If egg activation is independent of fertilization, does it mean both occur simultaneously or nearly so to avoid engaging the polyspermy block?

2. The discussion needs a completed final sentence. As it ends now, it seems incomplete.

3. It is unclear the connection to SOCE in the following statement, "Par2a as an essential regulator of IP3 levels at the zebrafish blastomere cleavage furrow, and acts with a known role of Store Operated Calcium Entry to replenish ER Ca++ levels (21)." The statement needs to be better explained, as SOCE was not mentioned previously, and the evidence in fish was not discussed.

---

## [Decision Letter · Decision Letter 2]

22 Mar 2025

Dear Dr Carney,

Thank you for your patience while we considered your revised manuscript entitled "Protease-activated receptor 2 links protease activity with calcium waves during egg activation and blastomere cleavage" for publication as a Research Article at PLOS Biology. This revised version of your manuscript has been evaluated by the PLOS Biology editors, the Academic Editor and the original reviewers.

Based on the reviews, we are likely to accept this manuscript for publication, provided you satisfactorily address the data and other policy-related requests stated below my signature. We will leave it up to you to follow the suggestions made by Reviewer 3.

In addition, we would like you to consider a suggestion to improve the title:

"Protease-mediated activation of Par2 elicits calcium waves during zebrafish egg activation and blastomere cleavage"

We expect to receive your revised manuscript within two weeks.

*Published Peer Review History*

*Press*

Sincerely,

Ines

--

Ines Alvarez-Garcia, PhD

Senior Editor

PLOS Biology

Fig. 1E, F, J; Fig. 2B, F, D; Fig. 3B, C, F, G; Fig. 4E, L; Fig. 5C, E, F; Fig. S1B, D, E; Fig. S3; Fig. S5G; Fig. S6E, J; Fig. S7C; Fig. S8 and Fig. S10

**In addition, please make sure the data you have deposited in GEO (GSE289416) is publicly available at this stage.

CODE POLICY

Reviewers' comments

Rev. 1:

I thank the authors for addressing my concerns. I have no additonal comments.

Rev. 2:

Thank you to the authors for their revisions. My questions have all been well addressed.

Rev. 3:

The authors have addressed my comments. The discussion has an underwhelming end. I suggest to move the considerations of the nature and mechanism of the protease to earlier in the section. Add a concluding statement that involves the notion of the conservation of the mechanisms and elaborate in the advantages of having such a an egg activation system.

---

## [Editor Report · Decision Letter 3]

25 Apr 2025

Dear Dr Carney,

Thank you for the submission of your revised Research Article entitled "Protease-mediated activation of Par2 elicits calcium waves during zebrafish egg activation and blastomere cleavage" for publication in PLOS Biology. On behalf of my colleagues and the Academic Editor, Carmen Williams, I am delighted to let you know that we can in principle accept your manuscript for publication, provided you address any remaining formatting and reporting issues. These will be detailed in an email you should receive within 2-3 business days from our colleagues in the journal operations team; no action is required from you until then. Please note that we will not be able to formally accept your manuscript and schedule it for publication until you have completed any requested changes.

PRESS

Sincerely, 

Ines

--

Ines Alvarez-Garcia, PhD

Senior Editor

PLOS Biology
